# The ability to sense the environment is heterogeneously distributed in cell populations

Andrew Goetz[1†], Hoda Akl[2†], Purushottam Dixit[1,3]*

[1]Department of Biomedical Engineering, Yale University, New Haven, United States; [2]Department of Physics, University of Florida, Gainesville, United States; [3]Systems Biology Institute, Yale University, West Haven, United States

**Abstract** Channel capacity of signaling networks quantifies their fidelity in sensing extracellular inputs. Low estimates of channel capacities for several mammalian signaling networks suggest that cells can barely detect the presence/absence of environmental signals. However, given the extensive heterogeneity and temporal stability of cell state variables, we hypothesize that the sensing ability itself may depend on the state of the cells. In this work, we present an information-theoretic framework to quantify the distribution of sensing abilities from single-cell data. Using data on two mammalian pathways, we show that sensing abilities are widely distributed in the population and most cells achieve better resolution of inputs compared to an '*average cell*'. We verify these predictions using live-cell imaging data on the IGFR/FoxO pathway. Importantly, we identify cell state variables that correlate with cells' sensing abilities. This information-theoretic framework will significantly improve our understanding of how cells sense in their environment.

*For correspondence:
purushottam.dixit@yale.edu

†These authors contributed equally to this work

Competing interest: The authors declare that no competing interests exist.

## eLife assessment

In this **valuable** article, the authors use an existing theoretical framework relying on information theory and maximum entropy inference in order to quantify how much information single cells can carry, taking into account their internal state. They reanalyze experimental data in this light. Despite some limitations of the data, the study **convincingly** highlights the difference between single-cell and population channel capacities. This result should be of interest to the quantitative biology community as it contributes to explaining why channel capacities are apparently low in cells.

## Introduction

In cell populations, there is a significant overlap in responses to environmental stimuli of differing strengths. This raises a fundamental question (*Levchenko and Nemenman, 2014*): do signaling networks in cells relay accurate information about their environment to take appropriate action? And if not, where along the signal transduction pathway is the information lost (*Rhee et al., 2012*; *ten Wolde et al., 2016*)? Mutual information (MI) quantifies the information content in an intracellular output about extracellular inputs. For an input $u$ (e.g., concentration of a ligand) and an output $x$ (e.g., intracellular species concentration, *Cheong et al., 2011*; *Selimkhanov et al., 2014*; *Suderman et al., 2017*) or a cellular phenotype (*Varennes et al., 2019*; *Mattingly et al., 2021*), the MI is defined as (*Cover and Thomas, 2006*)

$$I = \sum_{x,u} p\left(x|u\right) p\left(u\right) \log_2 \frac{p\left(x|u\right)}{\sum_{u'} p\left(x|u'\right) p\left(u'\right)} \tag{1}$$

Experimental single-cell methods such as flow cytometry (*Wu and Singh, 2012*), immunofluorescence (*Wu and Singh, 2012*), mass spectrometry (*Slavov, 2021*), or live-cell imaging (*Lemon and McDole, 2020*) allow us to estimate response histograms $p(x|u)$ across several inputs. Using these distributions, we can estimate the maximum of the MI (the channel capacity, CC) by optimizing *Equation 1* over input distributions $p(u)$. The CC quantifies fidelity of signal transduction (*Levchenko and Nemenman, 2014*; *Rhee et al., 2012*). For example, a CC of 1 bit implies that that the cells can *at best* distinguish between two input levels (e.g., presence versus absence), with higher CCs indicating that cells can resolve multiple input states. Importantly, CC can be used to identify bottlenecks in signaling (*Rhee et al., 2012*; *Hansen and O'Shea, 2015*).

The CC has been estimated for input–output relationships in several mammalian signaling networks (*Cheong et al., 2011*; *Selimkhanov et al., 2014*; *Suderman et al., 2017*; *Billing et al., 2019*; *Garner et al., 2016*; *Frick et al., 2017*). When the output is defined as levels of a single protein a fixed time, the CC was found to be surprisingly low, ~1–1.5 bits. These estimates have been improved by considering multidimensional outputs (*Selimkhanov et al., 2014*) or time-varying inputs (*Lee et al., 2021*). While these modifications led to somewhat higher CC estimates, the overall conclusion that cells know little about their environment remains well established. In contrast, significantly higher CC estimates have been found when the output at the level of cell population averages is considered (*Suderman et al., 2017*), suggesting that the only way to overcome low-sensing fidelity is population average response.

These previous calculations estimated one-channel capacity for all cells in a population, implicitly assuming that individual cells have similar sensing abilities. However, we now know that cells exhibit extensive heterogeneity in cell state variables (*Trapnell, 2015*; *Wollman, 2018*) such as abundances of key signaling proteins (*Meyer et al., 2012*; *Frankel et al., 2014*), mRNA abundances (*Chen et al., 2015*), and chromatin conformation (*Nagano et al., 2013*) and accessibility (*Buenrostro et al., 2015*). Notably, the time scale of fluctuations in these variables can be significantly slower than relevant signaling time scales (*Flusberg et al., 2013*; *Spencer et al., 2009*), sometimes extending across multiple generations (*Pleška et al., 2021*). Heterogeneity in cell state variables leads to a heterogeneity in response to extracellular cues, including chemotherapeutic drugs (*Flusberg et al., 2013*; *Spencer et al., 2009*; *Emert et al., 2021*), mitogens (*Matson and Cook, 2017*), hormones (*Norris et al., 2021*), chemotactic signals (*Varennes et al., 2019*; *Pleška et al., 2021*; *Moon et al., 2021*; *Waite et al., 2016*), and other electrical and chemical stimuli (*Clark and Campbell, 2019*; *Yao et al., 2016*). Therefore, we hypothesize that that the ability to sense the environment varies from cell to cell in populations in a cell state-dependent manner.

There is no conceptual framework to estimate the variation in sensing abilities in cell populations and its dependence on cell state variables. To that end, we introduce an information-theoretic quantity CeeMI: **Ce**ll stat**e**-dependent **M**utual **I**nformation. We show that using typically collected single-cell data and computational models of signaling networks, we can estimate the distribution $p_{\text{CeeMI}}(I)$ of single-cell signaling fidelities (single-cell mutual information values). We also show that we can identify cell state variables that make some cells better and others worse at sensing their environment.

Using an illustrative example, we show that in heterogeneous cell populations, estimates of mutual information that average over cell states can be significantly lower than the mutual information of signaling networks in typical cells. Next, using previously collected experimental data, we estimate $p_{\text{CeeMI}}(I)$ for two important mammalian signaling pathways (*Lewis Cantley and Sever, 2013*): the epidermal growth factor (EGF)/EGF receptor pathway and the insulin-like growth factor (IGF)/Forkhead Box protein O (FoxO) pathway. We show that while the cell state-agnostic CC estimates for both pathways are ~1 bit, most individual cells are predicted to be significantly better at resolving different inputs. Using live-cell imaging data for the IGF/FoxO pathway, we show that our estimate of variability in sensing abilities matches closely with experimental estimates. Importantly, for this pathway, we also verify our prediction that specific cell state variables dictate cells' sensing abilities. Finally, using a simple receptor/ligand model, we show how the time scales cell state dynamics affects cells' individuality and therefore sensing abilities. We believe that CeeMI will be an important tool to explore variability in cellular sensing abilities and in identifying what makes some cells better and others worse in sensing their environment.

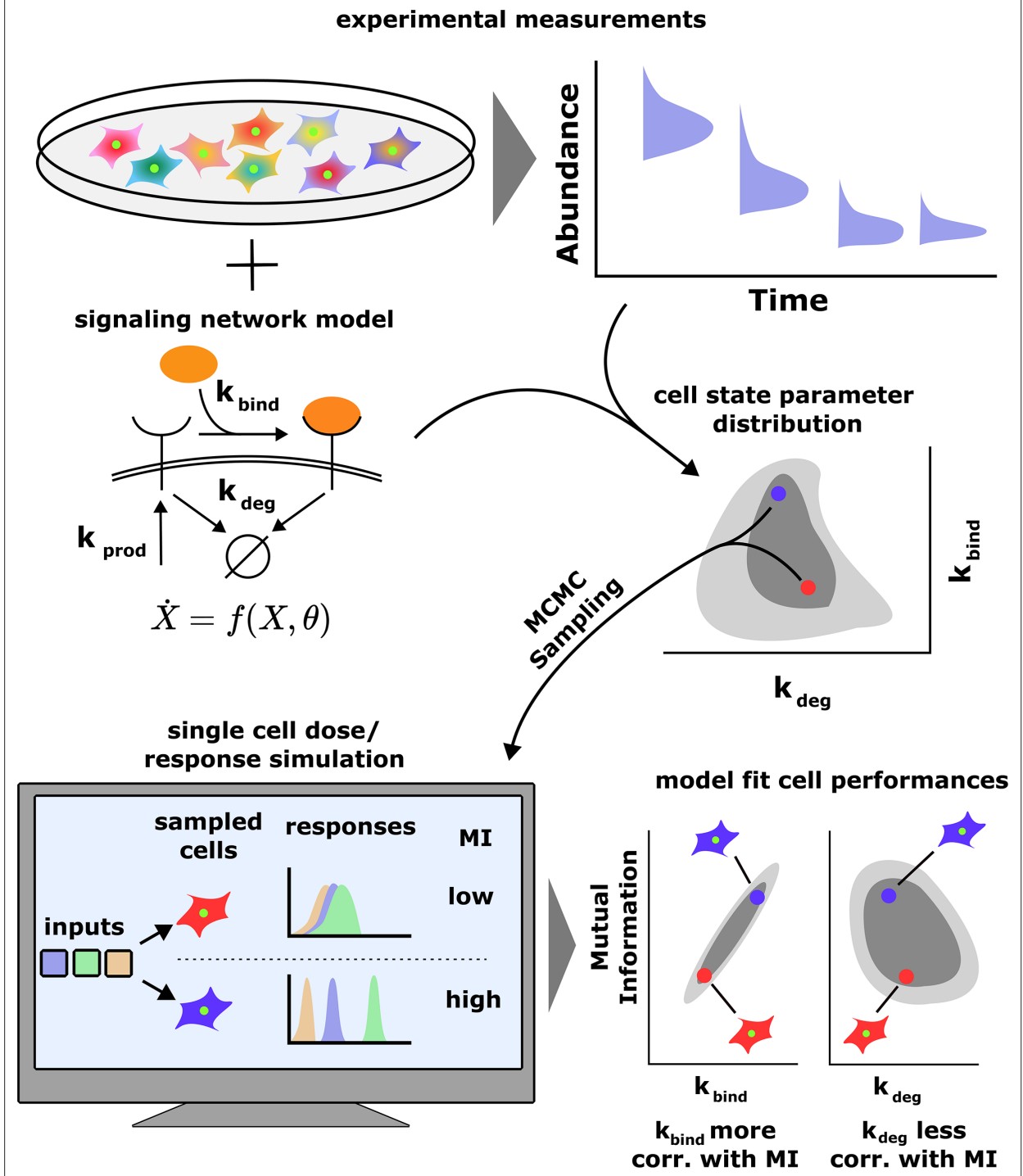

**Figure 1.** A schematic of our computational approach. Top: single-cell data across different input conditions and time points are integrated with a stochastic model of a signaling network using a previous developed maximum entropy approach leading to a distribution over signaling network parameters $p(\boldsymbol{\theta})$ (middle). Bottom: in silico cells are generated using the inferred parameter distribution and cell state-specific mutual information $I(\boldsymbol{\theta})$ and population distribution of cell performances $p_{\text{CeeMI}}(I)$ is estimated. The model also evaluates the correlation between cells' performance and biochemical parameters.

## Results

### Conditional mutual information models single cells as cell state-dependent channels

Consider a cell population where cells are characterized by state variables $\theta$ (*Figure 1*). These include abundances of signaling proteins and enzymes, epigenetic states, etc. We assume that cell states are temporally stable, that is, $\theta$ remains constant over a time scale that is longer than typical fluctuations in environmental inputs. Later, using a toy simple model, we explicitly study the effect of cell state dynamics on cells' ability to sense their environment.

We assume that cell state variables are distributed in the population according to a distribution $p(\theta)$. If $x$ denotes an output of choice (e.g., phosphorylation levels of one or more protein(s) at one or more time point(s)) and $u$ denotes the input (e.g., ligand concentration), the experimentally measured response distribution $p(x|u)$ can be decomposed as

$$p(x|u) = \int p(x|u, \theta) \, p(\theta) \, d\theta \tag{2}$$

where $p(x|u, \theta)$ is the distribution of the output $x$ conditioned on the input $u$ and cell state variables $\theta$. We note that in most cases the same cell cannot be probed multiple times. Consequently, $p(x|u, \theta)$ may not be experimentally accessible. However, it is conceptually well defined and mathematically accessible when interactions among molecules are specified (*Feinberg and Levchenko, 2023*).

Using $p(x|u, \theta)$, we can define the cell state-dependent mutual information $I(\theta)$ for a fixed input distribution $p(u)$ analogously to *Equation 1*:

$$I(\theta) = \sum_{x,u} p(x|u, \theta) \, p(u) \log_2 \frac{p(x|u, \theta)}{\sum_{u'} p(x|u', \theta) \, p(u')} \tag{3}$$

$I(\theta)$ quantifies individual cells' ability to sense their environment as a function of the cell state parameters $\theta$. The distribution $p_{\text{CeeMI}}(I)$ of single-cell sensing abilities is

$$p_{\text{CeeMI}}(I) = \int p(\theta) \, \delta(I - I(\theta)) \, d\theta \tag{4}$$

where $\delta(\cdot)$ is the Dirac delta function. We can also compute the joint distribution between the single-cell mutual information and any cell state variable of interest $\chi$ (e.g., abundance of cell surface receptors):

$$p_{\text{CeeMI}}(I, \chi) = \int p(\theta) \, \delta(I - I(\theta)) \, \delta(\chi - \chi(\theta)) \, d\theta \tag{5}$$

where $\chi(\theta)$ is the value of the biochemical parameter when cell state variables are fixed at $\theta$. The distributions in *Equation 5* quantify the interdependency between a cell's signaling fidelity $I(\theta)$ and cell-specific biochemical parameters $\chi(\theta)$. As we will see below, the distributions in *Equations 4* and *5* can be experimentally verified when appropriate measurements are available. Finally, we define the population average of the cell state-dependent mutual information:

$$I_{\text{Cee}} = \int I(\theta) \, p(\theta) \, d\theta \tag{6}$$

In information theory, $I_{\text{Cee}}$ is known as the conditional mutual information (*Cover and Thomas, 2006*) between the input $u$ and the output $x$ conditioned on $\theta$. If cell state variables remain constant and if distribution over cell states is independent of the input distribution, it can be shown that $I_{\text{Cee}} \geq I$ (Section 1 of Materials and Methods). It then follows that at least some cells in the population have better sensing abilities compared to the cell state-agnostic mutual information (*Equation 1*).

Since $I_{\text{Cee}}$ depends on the input distribution $p(u)$, we can find an optimal input distribution $p(u)$ that maximizes $I_{\text{Cee}}$ (Section 1 of Materials and Methods). Going forward, unless an input distribution is specified, the distributions $p_{\text{CeeMI}}(I)$ and $p_{\text{CeeMI}}(I, \chi)$ are discussed in the context of this optimal input distribution.

## Maximum entropy inference can estimate $p_{\text{CeeMI}}(I)$

Estimation of $p_{\text{CeeMI}}(I)$ requires decomposing the experimentally observed response $p(x|u)$ into cell-specific output distributions $p(x|u, \boldsymbol{\theta})$ and the distribution of cell state variables $p(\boldsymbol{\theta})$ (*Equations 3 and 4*). This problem is difficult to solve given that neither $p(x|u, \boldsymbol{\theta})$ nor $p(\boldsymbol{\theta})$ are accessible in typical experiments. However, for many signaling networks, stochastic biochemical models can be constructed to obtain the cell-specific output distribution $p(x|u, \boldsymbol{\theta})$. Here, $\boldsymbol{\theta}$ comprise protein concentrations and effective rates of enzymatic reactions and serve as a proxy for cell state variables. Given the experimentally measured cell state-averaged response $p(x|u)$ and the model-predicted cell-specific output distribution $p(x|u, \boldsymbol{\theta})$, we need a computational method to infer $p(\boldsymbol{\theta})$ (see *Equation 2*). The problem of inference of parameter heterogeneity is a non-trivial inverse problem (*Dixit et al., 2020*). While there are several proposed methods to solve this problem (reviewed in *Loos and Hasenauer, 2019*), most cannot efficiently infer $p(\boldsymbol{\theta})$ for signaling networks with even a moderate ($n \sim 10$) number of parameters. Here, we use our previously develop maximum entropy-based approach to infer $p(\boldsymbol{\theta})$ (*Dixit et al., 2020*). This way, we can estimate $p_{\text{CeeMI}}(I)$ using experimentally obtained cell state-agnostic response $p(x|u)$ and a stochastic biochemical model $p(x|u, \boldsymbol{\theta})$ of the underlying signaling network.

## An 'average cell' can discern much less than a typical cell about the environment

To illustrate the effect of heterogeneity of cell state variables on the cell state-agnostic estimate of mutual information (which we call $I_{\text{CSA}}$ from now onward), we consider a simple stochastic biochemical network of a receptor-ligand system. Details of the model and the calculations presented below can be found in Section 2 of Materials and Methods. Briefly, extracellular ligand $L$ binds to cell surface receptors $R$. Steady-state levels of the ligand-bound receptor $B$ are considered the output. The signaling network is parameterized by several cell state variables $\boldsymbol{\theta}$ such as receptor levels, rates of binding/unbinding, etc. For simplicity, we assume that only one variable, steady-state receptor level $R_0$ in the absence of the ligand, varies in the population. Calculations with variability in other parameters are presented in Section 2 of Materials and Methods.

In this population, cells' response $p(B|L, R_0)$ is distributed as a Poisson distribution whose mean is proportional to the cell state variable $R_0$ (Section 2 of Materials and Methods). That is, when all other parameters are fixed, a higher $R_0$ leads to lower noise (coefficient of variation). To calculate cell state-dependent mutual information (*Equation 3*), we assume that $p(L)$ is a gamma distribution. As expected, $I(R_0)$ (*Equation 3*) between the output $B$ and the input $L$ increases monotonically with $R_0$ (inset in *Figure 2A*). Moreover, given that $R_0$ varies in the population (also assumed to be a gamma distribution), the sensing ability varies in the population as well (*Figure 2A*). Notably, the average $I_{\text{Cee}}$ of $I(R_0)$ remains relatively robust to variation in $R_0$. At the same time, the traditional estimate $I_{\text{CSA}}$, which is the mutual information between the input $L$ and the cell state-agnostic population response $p(B|L) = \int p(B|L, R_0) p(R_0) dR_0$ (response of the 'average' cell, *Equations 1 and 2*), decreases as the population heterogeneity in $R_0$ increases. Importantly, $I_{\text{CSA}}$ is significantly lower than the sensing ability of most cells in the population (*Figure 2A*). This is because the overlap in the population response distributions is significantly larger than that in single-cell response distributions (*Figure 2B*). This simple example illustrates that the traditional mutual information estimates may severely underestimate cells' ability to resolve inputs, especially when cell state variables are heterogeneously distributed. Moreover, it is crucial to explicitly account for heterogeneity in cell state variables when estimating fidelity of cellular communication channels.

## Experimental verification of $p_{\text{CeeMI}}(I)$ using growth factor signaling networks

In cell populations, state variables that govern signaling dynamics such as protein levels (receptors, kinases, dephosphatases, etc.) (*Meyer et al., 2012*; *Frankel et al., 2014*), as well as effective rate constants such as endocytosis rates (*Kallenberger et al., 2017*), ligand binding rates (*Chung et al., 1997*), etc., vary from cell to cell. Therefore, we expect the downstream phenotype of environmental sensing to be widely distributed as well. To experimentally verify the computational prediction of the distribution $p_{\text{CeeMI}}(I)$ of sensing abilities, we need a system that allows us to approximate the cell state-specific response distribution $p(x|u, \boldsymbol{\theta})$. The IGF/FoxO pathway (*Lewis Cantley and Sever,*

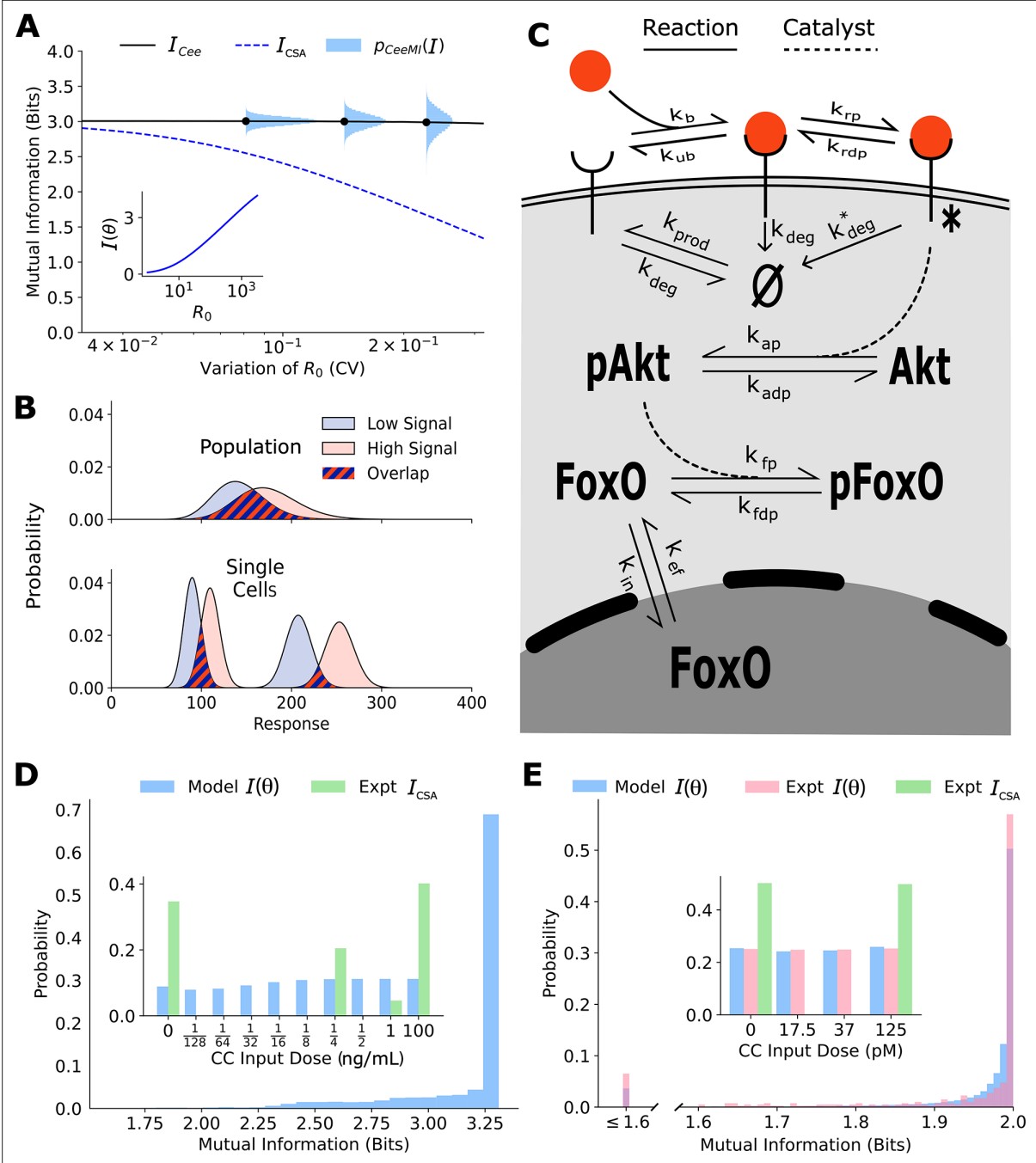

**Figure 2.** Channel capacities are broadly distributed in a population of cells. (**A**) The distribution of single-cell sensing abilities $p_{\text{CeeMI}}(I)$ (horizontal blue histograms) and its average $I_{Cee}$ plotted as a function of the coefficient of variation of the distribution of one cell state variable, the cell surface receptor number $R_0$. The dashed blue lines show the traditional cell state-averaged mutual information $I_{CSA}$ (*Equation 1*). The inset shows the dependence between cell state-specific mutual information $I(\theta)$ and cell state variable $R_0$. The input distribution $p(L)$ is assumed to be a gamma distribution. (**B**) A schematic showing the effect of heterogeneity in cell states on population-level response. Even when individual cells have little overlap in their responses to extracellular signal (bottom), the population-level responses could have significant overlap (top), leading to a low mutual information between cell state-averaged response and the input. (**C**) A combined schematic of the two growth factor pathways. Extracellular growth factor ligand (red circle) binds to cell surface receptors which are shuttled to and from the plasma membrane continuously. Ligand-bound receptors are phosphorylated and activate Akt. Phosphorylated Akt leads to phosphorylation of FoxO, which bars it from entering the nucleus. EGF/EGFR model is limited to the reactions on the plasma membrane. The corresponding cell state variables are given by $\boldsymbol{\theta} \equiv \left\{ k_{prod}, k_{bind}, k_{unbind}, k_p, k_{dp}, k_{deg}, k_{deg}^* \right\}$. The cell state variables for the IGF/FoxO model are given by $\boldsymbol{\theta} \equiv \left\{ k_{prod}, k_{bind}, k_{unbind}, k_p, k_{dp}, k_{deg}, k_{deg}^*, [Akt]_{total}, k_{ap}, k_{adp}, k_{in}, k_{ef}, k_{fp}, k_{fdp}, [FoxO]_{total} \right\}$. (**D**) The estimated distribution of single-cell mutual information values $p_{\text{CeeMI}}(I)$ for the EGF/EGFR pathway using a maximum entropy estimation of $\hat{p}(\boldsymbol{\theta})$.

*Figure 2 continued on next page*

*Figure 2 continued*

The inset shows the input distribution $p(u)$ corresponding to the maximum of the average $I_{Cee}$ of $p_{\text{CeeMI}}(I)$ (blue), along with the input distribution corresponding to the channel capacity of $I_{CSA}$ (green). **(E)** Same as **(D)** for the IGF/FoxO pathway. We additionally show the experimentally estimated $p_{\text{CeeMI}}(I)$ (pink).

The online version of this article includes the following figure supplement(s) for figure 2:

**Figure supplement 1.** Schematic of the toy model.

**Figure supplement 2.** Poisson distribution accurately fits numerically obtained distributions.

**Figure supplement 3.** Values of $I_{CSA}$ (*Equation S1*), $I_{Cee}$ (*Equation S3*), and $p_{CeeMI}(I)$ (y-axis) as a function of the degree of variability (coefficient of variation) in $k_{\text{deg}}$.

**Figure supplement 4.** Validating the accuracy of the moment closure approximation against histograms obtained using the Gillespie algorithm.

**Figure supplement 5.** Mean absolute relative error between predicted bin fractions and estimated bin fractions as a function of iteration number for the IGF/FoxO (panel A) and the EGF/EGFR pathway (panel B).

**Figure supplement 6.** A schematic of the experiment performed by Gross et al.

**Figure supplement 7.** Temporal stability of cell states.

*2013*) is an ideal system for these explorations for several reasons. First, following IGF stimulation, the transcription factor FoxO is pulled out of the nucleus. GFP-tagged variant of FoxO can be used to detect the dynamics of nuclear levels of FoxO at the single-cell level (*Gross et al., 2019*). Second, nuclear FoxO levels reach an approximate steady state within 30–45 min of constant IGF stimulation, with FoxO levels decreasing with increasing IGF dose (*Gross et al., 2019*). As a result, an approximate cell state-specific distribution $p(x|u,\theta)$ of steady-state levels of FoxO can be obtained by stimulating the same cell with successive IGF levels. Finally, the biochemical interactions in the IGF/FoxO are well studied (*Figure 2C*), allowing us to build a stochastic biochemical model based on previous computational efforts (*Wimmer et al., 2014*; *Lyashenko et al., 2020*) that fits the single-cell data accurately. Another system where $p_{\text{CeeMI}}(I)$ can in principle be verified is the EGF/EGFR pathway (*Figure 2C*). Here too, abundance of cell surface EGFR levels can be tracked in live cells following EGF stimulation (*Fortian and Sorkin, 2014*), allowing us to obtain cell state-specific response distribution $p(x|u,\theta)$. Below, we show estimates of $p_{\text{CeeMI}}(I)$ for both pathways and an experimental verification of our estimates for the IGF/FoxO pathway where live-cell imaging data was previously collected (*Gross et al., 2019*).

The details of the calculations presented below can be found in Materials and Methods (*Figure 2—figure supplement 7; Figure 2—figure supplement 6; Figure 2—figure supplement 5; Figure 2—figure supplement 3; Figure 2—figure supplement 2; Figure 2—figure supplement 1; Figure 2—figure supplements 4*). Briefly, we first constructed stochastic biochemical models for the two pathways based on known interactions (*Figure 2C*) and previous models (*Wimmer et al., 2014*; *Lyashenko et al., 2020*). The output for the EGFR pathway was defined as the steady-state levels of cell surface EGF receptors, and the output for the IGF/FoxO pathway was defined as the steady-state nuclear levels of the transcription factor FoxO. Using previously collected single-cell data on the two pathways (*Dixit et al., 2020*; *Gross et al., 2019*; *Lyashenko et al., 2020*) and our maximum entropy-based framework (*Dixit et al., 2020*), we estimated the distribution over parameters $p(\theta)$ for the two pathways (Section 3 of Materials and Methods). Using these inferred distributions, and the model-predicted cell state-specific response distribution $p(x|u,\theta)$, we could compute $p_{\text{CeeMI}}(I)$ for any specified input distribution $p(u)$. We choose the support of the input distribution as the ligand concentrations used in the experimental setup. The estimates shown in *Figure 2D and E* show $p_{\text{CeeMI}}(I)$ corresponding to the input distribution $p(u)$ that maximizes the conditional mutual information $I_{Cee}$ (see *Equation 6*). This input distribution is shown in the inset of *Figure 2D and E*.

Similar to the illustrative example (*Figure 2A*), there is a wide distribution of single-cell sensing fidelities in real populations (*Figure 2D and E*). Moreover, most cells are better sensors compared to the 'average cell', a cell whose response $p(x|u)$ is averaged over cell state variability which was estimated to be ~1 bit for both pathways. The cellular signaling fidelity skews toward the upper limit of 2 bits, which corresponds to the logarithm of the number of inputs used in the experiment. Indeed, as seen in the insets of *Figure 2D and E* and , the input distribution corresponding to the maximum of the cell state-agnostic mutual information $I_{CSA}$ is concentrated on the lowest and the highest input for

both pathways, indicating that cells may be able to detect only two input levels. In contrast, the input distribution corresponding to the maximum of $I_{\text{Cee}}$ is close to uniform, suggesting that individual cells can in fact resolve different ligand levels.

To verify our computational estimate of $p_{\text{CeeMI}}(I)$ for the IGF/FoxO pathway, we reanalyzed previously collected data wherein several cells were stimulated using successive IGF levels (*Gross et al., 2019*). The details of our calculations can be found in Section 4 of Materials and methods. Briefly, the cells reached an approximate steady state within 60 min of each stimulation and nuclear FoxO levels measured between 60 and 90 min were used to approximate an experimental cell state-specific response distribution $p(x|u, \boldsymbol{\theta})$. The distribution $p_{\text{CeeMI}}(I)$ was then obtained by maximizing the average mutual information $I_{\text{Cee}}$ (averaged over all cells) with respect to the input distribution. As seen in *Figure 2D and E*, the experimentally evaluated single-cell signaling fidelities match closely with our computational estimates. Moreover, as predicted using our computational analysis, individual cells in the population were significantly better at sensing than what is implied by the maximum of $I_{\text{CSA}}$. Indeed, the distribution of steady-state FoxO levels was found to be well resolved at the single-cell level as well (*Figure 2E*). Live-cell imaging data for the EGFR pathway was not available, and we leave it to future studies to validate our predictions.

Our calculations show that real cell populations comprise cells that have differing sensing fidelities, individual cells are significantly better at sensing their environment than what traditional estimates would indicate, and importantly, the CeeMI approach can accurately estimate the variation in signaling performances using readily collected experimental data and stochastic biochemical models. Notably, the variability in cell states and therefore the heterogeneity in sensing abilities are likely to be stable over time; the same cell's FoxO responses to the same input were found to have significantly less variation compared to the variation within the population (*Figure 2—figure supplement 7*).

## CeeMI identifies biochemical parameters that determine cells' ability to sense their environment

Like other phenotypes (*Frankel et al., 2014*) and signaling dynamics (*Meyer et al., 2012*; *Norris et al., 2021*), we expect that cells' ability to sense their environment depends on their state variables. For example, cells with faster endocytosis rates (*Kallenberger et al., 2017*) may integrate environmental

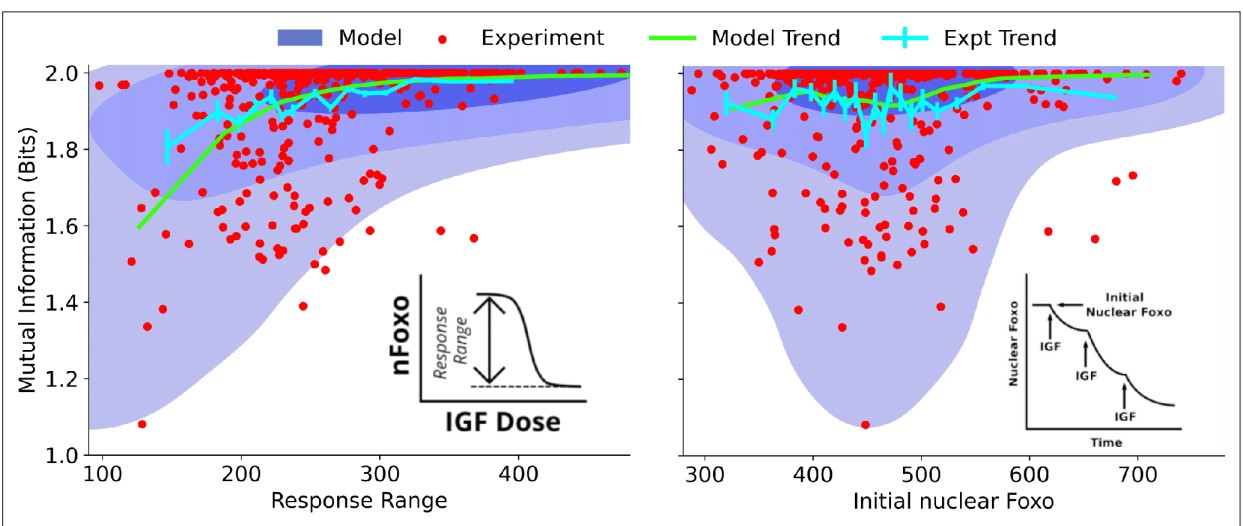

**Figure 3.** Dependence on cell state-dependent mutual information on biochemical parameters. Left: the joint distribution $p_{\text{CeeMI}}(I, \chi)$ of cell state-specific mutual information and biochemical parameter $\chi$ chosen to be the single-cell response range of nuclear FoxO levels (x-axis, see inset for a cartoon). The shaded blue regions are model predictions, and the green line is the model average. The darker shades represent higher probabilities. The red dots represent experimental cells. The cyan line represents experimental averages. Right: same as left, with biochemical parameter $\chi$ chosen to be steady-state nuclear Foxo levels in the absence of stimulation. The contours represent 1–10%, 10–50%, and 50–100% of the total probability mass (from faint to dark shading).

The online version of this article includes the following figure supplement(s) for figure 3:

**Figure supplement 1.** Joint distributions for additional biochemical parameters.

fluctuations with higher accuracy (*Lyashenko et al., 2020*). Or cells with higher receptor numbers (*Frankel et al., 2014*) will lead to a lowered relative noise in bound ligand concentration.

To systematically identify the variables that differentiate between cells' ability to sense the environment, we quantify the joint distribution $p_{\text{CeeMI}}(I, \chi)$ of single-cell signaling fidelity and biochemical state variables that take part in the signaling network. To test whether we can identify variables that are predictive of cellular fidelity, we estimated the joint distribution $p_{\text{CeeMI}}(I, \chi)$ for two variables that were experimentally accessible, response range of nuclear FoxO (*Figure 3*, left, see inset) and total nuclear FoxO levels prior to IGF stimulation (*Figure 3*, right, see inset). In both figures, the shaded regions show computational estimates of the joint probability distributions, and the red circles represent real cells. The green and the cyan trend lines represent computational and experimental binned averages.

One may expect that higher total nuclear FoxO levels could result in lower noise (coefficient of variation) and therefore better sensing abilities. However, we find that total nuclear FoxO levels only weakly correlate with cell state-dependent mutual information (correlation coefficient $r = 0.16$ for computational estimates and $r = 0.04$ for experimental data). In contrast, cell state-dependent mutual information depended strongly on the dynamic range of the response (correlation coefficient $r = 0.53$ for computational estimates and $r = 0.29$ for experimental data). Importantly, the model captured the observation that cells with a small response range had a variable sensing abilities while cells with a large response range all performed well in resolving extracellular IGF levels. Surprisingly, the total nuclear FoxO levels only weakly correlated with the cell-specific mutual information. In Section 5 of Materials and Methods (*Figure 3—figure supplement 1*), we show the predicted joint distributions $p_{\text{CeeMI}}(I, \chi)$ for several other biochemical variables that can potentially govern single cells' response to extracellular IGF stimuli. This way, CeeMI can be used to systematically identify cell state variables that differentiate between good sensors and bad sensors of the environmental stimuli.

## Time scale of stochastic dynamics of cell states dictates the divergence between state-specific and state-agnostic sensing abilities

A key limitation of the presented analysis is the assumption that cell state variables remain approximately constant over the time scale of typical environmental fluctuations. While many cell state variables change slowly, remaining roughly constant over the life spans of cells and beyond (*Flusberg et al., 2013*; *Spencer et al., 2009*), state changes may occur within the life span of a cell as well (*MacLean et al., 2018*). These dynamical changes can be accommodated easily in our calculations. Here, instead of fixing the cell state variables $\theta$, we can treat them as initial conditions and propagate them stochastically with their own dynamics. In the limit of very fast dynamics where individual cells rapidly transition cell states, we expect that the individuality of cells in a population vanishes and consequently cell state-specific mutual information for each cell will agree with traditional cell state-agnostic estimates of the channel capacity. In contrast, if the cell state dynamics are slow, our framework highlights differences between cells in the population.

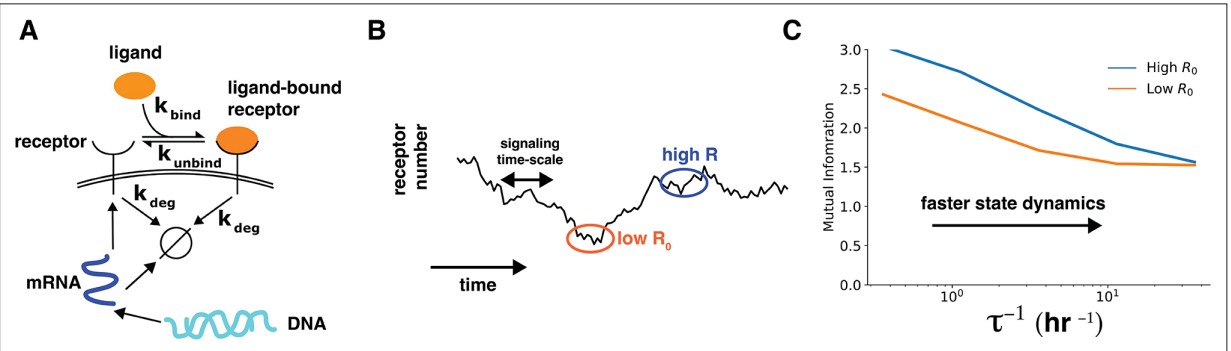

**Figure 4.** Cell state dynamics governs cell state-conditioned mutual information. (**A**) In a simple stochastic model, receptor mRNA is produced at a constant rate from the DNA and the translated into ligand-free receptors. The number of ligand-bound receptors after a short exposure to ligands is considered the output. (**B**) A schematic showing dynamics of receptor numbers when mRNA dynamics are slower compared to signaling time scales. (**C**) Conditioning on receptor numbers leads to differing abilities in sensing the environment when the time scale of mRNA dynamics is slow. In contrast, when the mRNA dynamics are fast (large $\tau^{-1}$), conditioning on cell state variables does not lead to difference in sensing abilities.

To elucidate the role of cell state dynamics, we built a simple model of ligand–receptor system (Section 6 of Materials and Methods, *Figure 4*). Briefly, the model included production and degradation of the receptor mRNA, translation of the receptor protein from the mRNA, and ligand binding to the receptor (*Figure 4A*). The number of ligand-free receptors and the number mRNA molecules were together considered the state variables. We tuned the mRNA dynamics by keeping the average mRNA number constant while simultaneously changing mRNA production and degradation rates. The time scale of mRNA dynamics is denoted by $\tau$.

When the mRNA dynamics are slower than translation, ligand binding, and receptor degradation, individual cells are effectively frozen around a particular receptor number (*Figure 4B*). The distribution of the number of ligand-bound receptors after a short-term ligand exposure reflects this frozen state. As a result, the cell state-conditioned mutual information depends strongly the cell state, with higher mutual information for cells with higher receptor numbers. In contrast, as the mRNA dynamics become faster, cells quickly lose the memory of cell state conditioning. As a result, the distribution of the number of ligand-bound receptors after a short-term ligand exposure reflects an ergodic averaging over the underlying mRNA dynamics. This averaging effectively washes out differences between cells (*Figure 4C*).

This example shows that when cells states change at time scales slower than signaling dynamics, cells in a population can be identified by their cell state and their ability to sense the environment can differ from one another.

## Discussion

Cell populations are characterized by heterogeneity in cell state variables (*Trapnell, 2015*; *Wollman, 2018*) that is responsible for important phenotypic differences and selective advantage, for example, in sensitivity to drugs (*Flusberg et al., 2013*; *Spencer et al., 2009*; *Emert et al., 2021*), response to chemotactic signals (*Mattingly and Emonet, 2022*), and following proliferation cues (*Matson and Cook, 2017*). Therefore, it is reasonable to expect that cells' ability to sense their environment depends on their state and is therefore variable across cells in a population. To quantify this heterogeneity, here, we developed a novel information-theoretic perspective that allowed us to quantify the distribution $p_{\mathrm{CeeMI}}(I)$ of single-cell sensing abilities using easily measurable single-cell data and stochastic models of signaling networks. We also quantified the joint distribution $p_{\mathrm{CeeMI}}(I, \chi)$ of cell-specific sensing ability and biochemical cell state variable. Importantly, using two growth factor pathways, we showed that individual cells in real cell populations were much better at sensing their environment compared to what is implied by the traditional estimate of channel capacities of signaling networks. Typical single-cell data are time stamped and do not give information about dynamics in the single cell. Moreover, these data do not provide information about dependence of cell-to-cell variability on cell state variables. Therefore, computational modeling of cellular trajectories from single-cell data and stochastic models is essential in deciphering mechanistic origins of heterogeneity in cellular phenotypes.

The approach presented here will be useful in identifying bottlenecks in signal transduction. Many cellular phenotypes such as chemotaxis (*Varennes et al., 2019*; *Mattingly et al., 2021*) and cell proliferation (*Gross and Rotwein, 2016*) exhibit a weak correlation between cellular outputs (e.g., directional alignment with chemical gradients) with the input (e.g., gradient strength), resulting in a low channel capacity even for individual cells. In such cases, it is important to understand where exactly along the information transduction pathway is the information about the gradient is lost. If traditional calculations are pursued, for example, for movement of mammalian cells under growth factor gradients (*Varennes et al., 2019*; *Moon et al., 2021*), one may conclude that the information loss likely happens right at the receptor level (*Figure 2D*). In contrast, CeeMI will allow us to disentangle the effect of cell state heterogeneity and noisy cellular response to precisely pinpoint intracellular signaling nodes that are responsible for signal corruption.

How do we contrast our results with previous low estimates of channel capacities? There is extensive population heterogeneity in cell state variables and this heterogeneity often is stable over time. Nonetheless, this heterogeneity arises from stochasticity in underlying processes including dynamics of epigenetic transitions and gene expression. When one specifies state variables (*Wollman, 2018*) $\theta$, one effectively conditions on the stochastic trajectory of the underlying dynamics to a specific narrow

window (**Figure 4**). This extra conditioning narrows the cells' response distributions, revealing the heterogeneous nature of cell populations (**Wollman, 2018**).

In summary, we showed that like other phenotypes the ability to sense the environment is itself heterogeneously distributed in a cell population. Moreover, we also showed that when conditioned on cell state variables, mammalian cells appear to be significantly better at sensing their environment than what traditional mutual information calculation suggests. Finally, we showed that we could identify cell state variables that made some cells better sensors compared to others. We believe that CeeMI will be an important framework in quantifying fidelity of input/output relationships in heterogeneous cell populations.

## Materials and methods

Scripts used to generate mutual information results as well the experimental data used to train our models can be obtained at https://github.com/adgoetz186/Cell_signalling_information (copy archived at **Goetz, 2024**). Script for maximum entropy inference of cellular parameters can be obtained at https://github.com/hodaakl/MaxEnt (copy archived at **Akl, 2024**).

The description of the methods is organized as follows: Section 1 describes basic information theoretical concepts and our numerical approach to estimate mutual information and channel capacity. Section 2 describes the in silico toy model. Section 3 describes the maximum entropy approach. Section 4 describes numerical methods for quantification of mutual information and channel capacity using live-cell imagining data. Section 5 shows the joint distribution of mutual information and biochemical parameters. Section 6 describes the methods to examine the role of cell state dynamics on sensing ability.

### 1. Information theory primer

Here, we give a brief description of information theoretic quantities used in the article. The readers are referred to a textbook (**Cover and Thomas, 2006**) for a detailed discussion.

A communication channel (e.g., a signaling network) is an input/output relationship between two random variables: the input $U$ (say, a ligand concentration) and the response $R$ (e.g., levels of some intracellular protein). The mutual information (MI) between $U$ and $R$ is the reduction in the uncertainty about $U$ due to the access of the outcome of $R$. The MI is defined as

$$I \equiv I(U;R) = \sum_{r \in R, u \in U} p(r|u) p(u) \log_2 \frac{p(r|u)}{\sum_{u'} p(r|u') p(u')} \tag{S1}$$

where the choice of two in the base of the logarithm gives the value of information in bits. The summation is replaced by integrals when considering continuous random variables. In the context of cell signaling, we will use the notation $I_{CSA}$ (instead of $I$ in **Equation S1**) to denote the cell state-agnostic mutual information, which is the mutual information between the response distribution $p(r|u)$ and the input distribution $p(u)$.

We also consider a more nuanced situation where instead of a single channel we have a family of channels whose states are characterized by a random variable $\Theta$. The channel state $\theta \in \Theta$ uniquely determines the probabilistic response relationship $p(r|u, \theta)$ between $U$ and $R$ when $\Theta$ is fixed. Similar to **Equation S1**, we can define the MI between $U$ and $R$ conditioned for a specific realization $\theta \in \Theta$

$$I(\theta) = \sum_{r \in R, u \in U} p(r|u, \theta) p(u) \log_2 \frac{p(r|u, \theta)}{\sum_{u'} p(r|u', \theta) p(u')} \tag{S2}$$

The average of $I(\theta)$ over $p(\theta)$ is traditionally known as the conditional mutual information (**Cover and Thomas, 2006**). We define the **Ce**ll stat**e**-dependent mutual information $I_{Cee}$

$$I_{Cee} \equiv I(U;R|\Theta) = \sum_{\theta \in \Theta} I(\theta) p(\theta) \tag{S3}$$

The channel capacity (CC) is a measure of the optimal performance of the signaling network with respect to the distribution of $U$. It can be defined for both $I$ and $I_{Cee}$ as follows:

$$CC_I = \max_{p(u)} I(U;R) \; or \; CC_{I_{Cee}} = \max_{p(u)} I(U;R|\Theta) \tag{S4}$$

The difference between $I$ and $I_{Cee}$ is called the interaction information $I(U;R;\Theta)$. The interaction information can be simplified as

$$I(U;R;\Theta) = I(U;R) - I(U;R|\Theta) = I(U;\Theta) - I(U;\Theta|R) = -I(U;\Theta|R) \leq 0 \tag{S5}$$

where $I(U;\Theta) = 0$ follows from the statistical independence of the input signal $U$ and channel state $\Theta$. **Equation S5** shows that $I_{Cee} > I$ as long as $I(U;\Theta) = 0$. We note that this may not be true in general. However, it is true in the context of cellular signaling networks where the inputs are chosen by the experimentalists while the cell states are an inherent property of the cell population. It is trivial to extend this argument to the channel capacities of both terms.

## Numerical estimation of mutual information

Evaluating the mutual information between an input and an output (**Equation S1** and **S2**) requires numerical integration over the input and the output distribution. We limit the input distributions to a finite support, specifically to the ligand concentrations that were used in the experimental setup. These are $L = [0, 0.0078, 0.01, 0.03, 0.06, 0.125, 0.25, 0.5, 1, 100]$ ng/mL for the EGF/EGFR pathway and $L = [0, 17.5, 37.5, 125]$ pM for the IGF/FoxO pathway.

The numerical integration in **Equation S1** and **S2** requires summing over probabilities of all possible responses and inputs. When the response distributions are approximated using histograms (either from experimental data or from Markov chain Monte Carlo simulations of a model), the summation can be error prone. To avoid this, we assume that the responses are distributed according to a gamma distribution (see **Figure 2—figure supplement 4**). The gamma distribution was chosen here and in several other places because it has been shown to accurately approximate real distributions of protein/mRNA abundances (**Taniguchi et al., 2010**).

To speed up the calculations, inspired by previous approaches (**Rhee et al., 2012**), we use a binning strategy. Specifically, we bin the response distribution using a constant bin width. The bin width was chosen to be equal to 5% of the smallest interquartile range across response distributions corresponding to all considered inputs. The binning procedure requires truncating the response distributions to a finite support. We ensured that our binning captured at least 99.95% of the entire mass of the distribution. The same strategy was used to obtain $I_{CSA}$ in **Equation S1** and $I(\theta)$ in **Equation S2**. The only difference in computing $I(\theta)$ compared to $I_{CSA}$ was that the response distribution $p(r|u, \theta)$ was obtained computationally using a stochastic differential equation model with network parameters fixed at $\theta$. Samples from the joint distribution $p_{CeeMI}(I, \chi)$ where $\chi$ is a cell state variable were obtained by sampling cell state variables $\theta$ from $p(\theta)$ (see Section 3) and simultaneously evaluating $I(\theta)$ and $\chi(\theta)$.

Once the mutual information could be obtained numerically for a given input distribution $p(u)$, the corresponding channel capacity, the maximum of the mutual information over all possible input distributions, can be obtained by solving the following optimization problem:

$$\max_{p(u)} I_{CSA} \; s.t. \; \sum p(u) = 1 \; and \; p(u) \geq 0 \tag{S6}$$

The optimization problem was solved via a trust region constrained algorithm (**Byrd et al., 1999**) using the SciPy optimization library (**Virtanen et al., 2020**). When the distribution over cell state variables $p(\theta)$ is available, we can estimate $I_{Cee}$ as the average $\langle I(\theta) \rangle_\theta$ (**Equation S3**). The optimum value of $I_{Cee}$ over all input distributions serves as a cell state-dependent analogue to the channel capacity of $I_{CSA}$ and is obtained by solve a similar optimization problem:

$$\max_{p(u)} I_{Cee} \; s.t. \; \sum p(u) = 1 \; and \; p(u) \geq 0 \tag{S7}$$

The procedure needed to evaluate single-cell mutual information values using live-cell imaging data on the IGF/FoxO pathway is described in Section 4.

## 2. Description of the in silico toy model

The in silico cell receptor/ligand system comprises two components (unbound receptors and bound receptors) and five reactions (*Figure 2—figure supplement 1*).

In the model, receptors are constantly shuttled to the membrane and removed from the membrane and degraded. The extracellular ligand (concentration denoted by $L$) binds to cell surface receptors. We assume that the ligand concentration is kept constant in the environment. Steady-state abundance of ligand-bound receptor (denoted by $B$) is taken as the output of the system. At steady state, the mean field equations for the average species level are

$$k_{\text{prod}} - k_{\text{bind}}LR + k_{\text{unbind}}B - k_{\text{deg}}R = 0 \tag{S8}$$

$$k_{\text{bind}}LR - k_{\text{unbind}}B - k_{\text{deg}}B = 0 \tag{S9}$$

Solving for steady state, at the single-cell level, the mean number of bound receptors is given by

$$\mu_B = R_0 \frac{Lk_{\text{bind}}}{Lk_{\text{bind}} + k_{\text{deg}} + k_{\text{unbind}}} \tag{S10}$$

In *Equation S10*, $k_{\text{bind}}$ is the ligand binding rate, $k_{\text{unbind}}$ is the ligand unbinding rate, $k_{\text{deg}}$ is the degradation rate, and $R_0$ is the average value of the receptor level in the absence of the ligand. The bound receptor levels are Poisson distributed with a mean given by *Equation S10* (*Figure 2—figure supplement 2*).

Using this toy network, we created two in silico cell populations. In both populations, we fixed $k_{\text{bind}} = 1 \ sec^{-1} a.u.^{-1}$ and $k_{\text{unbind}} = 10 \ sec^{-1}$. In the first population, every parameter was kept constant across cells except for the cell surface receptor levels $R_0$. In the second population, every parameter was kept constant across cells except for the receptor degradation rate $k_{\text{deg}}$. In the first population (when $R_0$ was varied), we fixed $k_{\text{deg}} = 5 sec^{-1}$. In the second population (when $k_{\text{deg}}$ was varied), we fixed $R_0 = 50$ molecules/cell. The parameter that varied across cells was assumed to be distributed according to a gamma distribution. For the two populations, we kept fixed mean value of the variable parameter to be $\langle R_0 \rangle = 500$ molecules/cell and $\langle k_{\text{deg}} \rangle = 5 \ sec^{-1}$. We varied coefficients of variation for both populations between $CV = 10^{-1.5}$ and $CV = 10^{-0.5}$.

### Obtaining $I_{CSA}$ for the toy network

*Equation 1* shows that calculation of MI in a cell state-agnostic manner requires the knowledge of the cell state-averaged response distribution $p(B|L)$ and the distribution of inputs $p(L)$. In our calculations, we restrict $p(L)$ to be a discrete version of the gamma distribution obtained by equal percentile binning of a gamma distribution. The mean of the gamma distribution was taken to be 10, the coefficient of variation 1, and the number of bins was 25. The discretization step was not essential but was taken to simplify the calculations by making all variables discrete.

We assumed that $p(B|L)$, which is obtained by averaging over the cell state variables, was distributed as a negative binomial distribution (*Figure 2—figure supplement 2*). We estimated its first two moments by numerically averaging the first two moments of the single-cell response distribution $p(B|L, \boldsymbol{\theta})$ (Poisson distribution with mean given by *Equation S10*) according to the population distribution $p(\boldsymbol{\theta})$ of the variable parameter(s) $\boldsymbol{\theta}$ ($\boldsymbol{\theta} \equiv R_0$ for the first cell population and $\boldsymbol{\theta} \equiv k_{\text{deg}}$ for the second cell population). As mentioned above, $p(\boldsymbol{\theta})$ was assumed to be a gamma distribution whose coefficient was systematically varied. Using the first two moments, we inferred the parameters for the negative binomial distribution.

Once the population-level response $p(B|L)$ is obtained and the input distribution $p(L)$ is fixed, we can calculate $I_{CSA}$ using *Equation 1*. The values of $I_{CSA}$ for the population with variable $R_0$ are given in *Figure 2C* while those for the population with variable $k_{\text{deg}}$ are given in *Figure 2—figure supplement 3*.

### Obtaining $p_{CeeMI}(I)$ and $I_{Cee}$ for the toy receptor network

As indicated by *Equations 4* and *6*, calculation of the distribution of cell state-dependent mutual information values requires the cell state-specific response distribution $p(B|L, \boldsymbol{\theta})$ and the distribution $p(\boldsymbol{\theta})$ of cell states. In the toy model, $p(B|L, \boldsymbol{\theta})$ is modeled as a Poisson distribution with a mean given by *Equation S8* and $p(\boldsymbol{\theta})$ is assumed to be gamma distributed in the variable parameter.

Using $p\left(B|L,\boldsymbol{\theta}\right)$ and gamma distributed input distribution $p\left(L\right)$, we obtain cell state-specific mutual information $I\left(\boldsymbol{\theta}\right)$ using *Equation S2*. We find $p_{CeeMI}\left(I\right)$ by sampling multiple values of $\boldsymbol{\theta}$ ($R_0$ for the first population and $k_{\mathrm{deg}}$ for the second population). $I_{Cee}$ is simply the average of $p_{CeeMI}\left(I\right)$. $I_{Cee}$ and $p_{CeeMI}\left(I\right)$ for the population with variable $R_0$ are given in *Figure 2C*. *Figure 2—figure supplement 3* shows the same for the population with variable $k_{\mathrm{deg}}$ .

## 3. Maximum entropy inference of cell state variability

Using experimentally collected single-cell data on heterogeneity in protein abundances, we estimate the distribution over cell state variables (biochemical parameters) using the maximum entropy approach. Below, we first describe the data that was used in our analysis. Next, we briefly discuss the maximum entropy approach.

### EGF/EGFR pathway, data, and model

We used previously collected single-cell data on cell surface EGFR levels (*Dixit et al., 2020*; *Lyashenko et al., 2020*). Briefly, MCF10A cells were stimulated with 10 different extracellular EGF levels ranging between 0 ng/mL and 100 ng/mL. Cell surface EGFR levels were measured in ~7000 cells for each ligand concentration after 3 hr of continuous EGF stimulation. The data was measured in arbitrary fluorescence units. To convert the data to the units of number of receptors per cell, we used a mean number of cell surface receptors $R = 2.5 \times 10^5$ for MCF10A cells (*Shi et al., 2016*). The population mean of the experimentally measured steady-state receptor count distribution in the absence of the ligand was matched to this number. Receptor count data at every other ligand concentration was scaled appropriately.

Using a previously validated model (*Dixit et al., 2020*; *Lyashenko et al., 2020*), we constructed a simplified model of the EGF/EGFR pathway. Specifically, we incorporated ligand binding to receptor, receptor activation, and preferential endocytosis of activated receptors. To keep our model simple, we did not incorporate receptor dimerization and oligomerization following ligand binding. Notably, evidence suggests that oligomers may be preformed in the absence of the ligand as well, which could make them effective monomers. Finally, we assumed that the extracellular ligand concentration was kept constant. The model was represented by the following reaction network.

$$\phi \overset{k_{\mathrm{prod}}}{\to} R \tag{S11}$$

$$R \overset{L\cdot k_{\mathrm{bind}}}{\to} B, \quad B \overset{k_{\mathrm{unbind}}}{\to} L+R \tag{S12}$$

$$B \overset{k_{\mathrm{p}}}{\to} P, \quad P \overset{k_{\mathrm{dp}}}{\to} B \tag{S13}$$

$$R \overset{k_{\mathrm{deg}}}{\to} \phi, \quad B \overset{k_{\mathrm{deg}}}{\to} \phi, \quad P \overset{k_{\mathrm{deg}}^*}{\to} \phi \tag{S14}$$

In *Equations S11–S14*, $R$ is the level of free receptor, $B$ is the level of ligand-bound receptor, and $P$ is the level of phosphorylated receptor. The total number of receptors is given by $R_T = R + B + P$. Cell state variables $\boldsymbol{\theta}$ comprised $\boldsymbol{\theta} \equiv \left\{ k_{prod},\ k_{bind},\ k_{unbind},\ k_p,\ k_{dp},\ k_{deg},\ k_{deg}^* \right\}$ .

### IGF/FoxO pathway, data, and model

We used previously collected single-cell data on nuclear FoxO levels following continuous IGF stimulation in HeLa cells (*Gross et al., 2019*). Briefly, cells were continuously stimulated with IGF, and nuclear FoxO levels were measured using GFP-tagged FoxO using live-cell imaging every 3 min for 90 min. To convert the arbitrary fluorescence units to units of copies of FoxO per cell, we first removed the background fluorescence intensity. Then, we used the previously estimated total FoxO levels in HeLa cells (*Bekker-Jensen et al., 2017*) (~710 molecules/cell) and the nuclear-to-cytoplasmic ratio in the absence of stimulation (*Wimmer et al., 2014*) (2/3 of the total in the nucleus). There was a small disagreement in mean nuclear FoxO levels in the absence of stimulation across different experiments. To remove this artifact and to start all experiments with the average nuclear FoxO levels, we offset individual experiments such that the mean nuclear FoxO was identical across all experiments.

Similar to the EGF/EGFR pathway, using a previously validated model (*Wimmer et al., 2014*), we constructed a simplified model of the IGF/FoxO pathway. Specifically, we incorporated ligand binding to IGF receptor, receptor activation, activation of Akt, and Akt-driven phosphorylation of FoxO. Phosphorylated FoxO was prohibited from entering the nucleus. Finally, we assumed that the

extracellular ligand concentration was kept constant. The model was represented by the following reaction network.

$$\phi \overset{k_{\text{prod}}}{\to} R \tag{S15}$$

$$R \overset{L \cdot k_{\text{bind}}}{\to} B, \quad B \overset{k_{\text{unbind}}}{\to} L + R \tag{S16}$$

$$B \overset{k_{\text{p}}}{\to} P, \quad P \overset{k_{\text{dp}}}{\to} B \tag{S17}$$

$$R \overset{k_{\text{deg}}}{\to} \phi, \quad B \overset{k_{\text{deg}}}{\to} \phi, \quad P \overset{k_{\text{deg}}}{\to} \phi \tag{S18}$$

$$Akt \overset{P \cdot k_{\text{ap}}}{\to} pAkt, \quad pAkt \overset{k_{\text{adp}}}{\to} Akt \tag{S19}$$

$$FoxO_c \overset{k_{\text{in}}}{\to} FoxO_n, \quad FoxO_n \overset{k_{\text{ef}}}{\to} FoxO_c \tag{S20}$$

$$FoxO_c \overset{pAkt \cdot k_{fp}}{\to} pFoxO_c, \quad pFoxO_c \overset{k_{\text{fdp}}}{\to} FoxO_c \tag{S21}$$

In *Equations S15–S21*, $R$ is the level of free IGF receptor, $B$ is the level of ligand-bound receptor, and $P$ is the level of phosphorylated receptor. $pAkt$ is phosphorylated Akt, $pFoxO_c$ is cytoplasmic phosphorylated FoxO, $FoxO_c$ is cytoplasmic unphosphorylated FoxO, and $FoxO_n$ is nuclear unphosphorylated FoxO. Cell state variables $\boldsymbol{\theta}$ comprised $\boldsymbol{\theta} \equiv \left\{ k_{prod}, k_{bind}, k_{unbind}, k_p, k_{dp}, k_{deg}, k_{deg}^*, [Akt]_{total}, k_{ap}, k_{adp}, k_{in}, k_{ef}, k_{fp}, k_{fdp}, [FoxO]_{total} \right\}$.

## Inference of model parameters

We assume that cells in a population can be assigned a cell-specific state denoted by a state vector $\boldsymbol{\theta}$ that comprises biochemical parameters relevant to the modeled signaling network. The population variability in cell state parameters is represented by the joint probability density $p(\boldsymbol{\theta})$. Typically, $p(\boldsymbol{\theta})$ is not experimentally accessible. Therefore, we infer it using a previously developed technique called MEDIRIAN (maximum entropy-based framework for inference of heterogeneity in dynamics of signaling networks) (*Dixit et al., 2020*). MERIDIAN infers the maximum entropy distribution $p(\boldsymbol{\theta})$ that reproduces a set of averages computed from experimental single-cell measurements.

MERIDIAN requires a mechanistic model of the signaling network that can predict cell's response to extracellular perturbation (e.g., ligand) and user-specified population averages computed from experimental data. For the EGF/EGFR and IGF/FoxO networks, we used stochastic biochemical models described by *Equations S11–S21*, respectively. We use the moment closure approximation (*Gillespie, 2009*) to approximate the single-cell distributions using the first two moments (see below). The differential equations for the pathways can be found on the GitHub.

Now, we briefly describe the MERIDIAN approach. The entropy of any distribution $p(\boldsymbol{\theta})$ is given by

$$S = -\int p(\boldsymbol{\theta}) \log p(\boldsymbol{\theta}) \, d\theta \tag{S22}$$

In MERIDIAN, we find $p(\boldsymbol{\theta})$ that maximizes $S$ while requiring it to reproduce a set of average constraints evaluated using experimental data. Following *Dixit et al., 2020*, entropy maximized $p(\boldsymbol{\theta})$ is given by the Gibbs–Boltzmann distribution

$$p(\boldsymbol{\theta}) = \frac{1}{\Omega} \exp\left( -\sum_m \lambda_m \psi_m(\boldsymbol{\theta}) \right) \tag{S23}$$

where $\lambda_m$ are the Lagrange multiplier corresponding to the $m$th constraint, $\psi_m(\boldsymbol{\theta})$ is the quantity whose average is constrained, and a $\Omega$ is the normalization constant.

Given a set of constraints (see below), the Lagrange multipliers can be numerically tuned such that the predictions from the distribution $p(\boldsymbol{\theta})$ match their experimental value. We optimize the Lagrange multipliers using gradient-based search (*Equation S24*) using the ADAM algorithm (*Kingma, 2014*). The gradients for minimizing a Lagrangian cost function are given by *Dixit et al., 2020*.

$$L = \log \Omega + \sum_m \lambda_m R_m$$

$$\frac{\partial L}{\partial \lambda_m} = R_m - \langle \psi_m(\boldsymbol{\theta}) \rangle_{\boldsymbol{\theta}} \tag{S24}$$

In *Equation S24*, $\langle \psi_m(\boldsymbol{\theta}) \rangle_{\boldsymbol{\theta}}$ denotes the ensemble average computed using $p(\boldsymbol{\theta})$ and $R_m$ are the corresponding measurements. We stop the iterative procedure when the mean absolute relative error $\frac{1}{M} \sum_m \frac{|R_m - \langle \psi_m(\boldsymbol{\theta}) \rangle_{\boldsymbol{\theta}}|}{R_m}$ reaches a predefined value.

The predictions from the Max Ent model depend on the choice of the experimental constraints. To ensure that the constraints represent the entire range of single-cell behaviors, we opt for percentile constraints. Specifically, for experimentally collected single-cell data for a given condition (ligand dose, time point, etc.), we first approximate the single-cell histogram using a gamma distribution. Then, we identify abundances that represent 10th–90th percentiles of this distribution. The fraction of cells belonging to each of these percentile windows is exactly 10%. These become our experimental constraints $R_{em} = 0.1$. Here, 'e' denotes the experimental condition (ligand dose, time point, etc.) and $m \in [1, 10]$ denotes the percentile window.

The corresponding model predictions of the fraction of cells in a given percentile window $\langle \psi_{em}(\boldsymbol{\theta}) \rangle_{\boldsymbol{\theta}}$ are given by

$$\langle \psi_{em}(\boldsymbol{\theta}) \rangle_{\boldsymbol{\theta}} = \int p(\boldsymbol{\theta}) \, \psi_{em}(\boldsymbol{\theta}) \, d\boldsymbol{\theta} \tag{S25}$$

where

$$\psi_{em}(\boldsymbol{\theta}) = \int_{l_{em}}^{u_{em}} p_e(r|\boldsymbol{\theta}) \, dr \tag{S26}$$

In *Equation S26*, $p_e(r|\boldsymbol{\theta})$ is the model-predicted single-cell distribution of responses (surface EGFR levels or nuclear FoxO levels) for experimental condition $e$. The integration bounds $l_{em}$ and $u_{em}$ represent the lower and upper bounds of the $m$th percentile window. The distribution $p_e(r|\boldsymbol{\theta})$ in principle can be approximated by several runs of an explicit simulation using Gillespie's algorithm (*Billing et al., 2019*). This may prove to be computationally expensive, especially when sampling through multiple parameter sets $\boldsymbol{\theta}$. Therefore, we resort to moment closure techniques (*Gillespie, 2009*) to approximate $p_e(r|\boldsymbol{\theta})$ as a gamma distribution. We used a previously developed package called MOCA (moment closure analysis) (*Schnoerr et al., 2015*). We used a Gaussian moment closure to obtain the first and the second moments of the distributions. The moment closure approximation was quite accurate compared to the explicit Gillespie simulation and allowed us to rapidly predict single cell response distributions without performing multiple calculations (*Figure 2—figure supplement 4*).

The averages in *Equation S25* cannot be computed analytically. We therefore resort to Markov chain Monte Carlo techniques to approximate them. Briefly, for a fixed set of Lagrange multipliers, we start 150 parallel MCMC chains in the parameter space with a starting point chosen randomly from the previous iteration. Each step in the MCMC calculation attempted to change between 1 and 5 parameters. The step size for the change was a uniform random number whose maximum was 10% of the parameter bounds for individual parameters. Each MCMC chain was run for approximately for $\sim 10^4$ steps for the EGF/EGFR pathway and $\sim 2.5 \times 10^5$ for the IGF/FoxO pathway. The first 1000 steps were discarded, and parameter sets were stored every 50th step after that.

We used ADAM to optimize the Lagrange multipliers (*Figure 2—figure supplement 5*). The hyperparameters for ADAM were as follows: the exponential decay rate for the first moment estimates was set to 0.8. The exponential decay rate for the second-moment estimates was set to 0.999, the step size was 0.1. This procedure led to a decrease in the relative error. Using the final set of Lagrange multipliers, we sampled parameter sets that represented an in silico cell population. This population was used for further analysis (see below).

## 4. Numerical quantification of mutual information and channel capacity using live-cell imaging data

Here, we describe how we estimated cell state-specific mutual information from live-cell imaging data. The IGF/FoxO pathway reaches an approximate steady state within 30–45 min after the introduction of the IGF ligand. We used data collected on HeLa cells where cells were treated with subsequent doses of IGF every 90 min to approximate the single-cell response to multiple IGF levels. We approximated the single-cell steady-state nuclear FoxO distributions as gamma distributions and estimated their means and variances from data collected between 60 and 90 min of IGF stimulation

(*Figure 2—figure supplement 6*). Using these first two moments, we approximated the single-cell response distribution $p(r|u, \theta)$ as a gamma distribution. The mutual information at the single-cell level and the population average cell state-specific mutual information $I_{Cee}$ were obtained using these response distributions. For $I_{CSA}$, the first two moments of the single-cell response distributions were averaged to obtain the population-level means and variances. This, along with the assumption that the responses were gamma distributions, provided the cell state-agnostic response distribution $p(r|u)$ from which $I_{CSA}$ was obtained.

To verify that cell states are indeed conserved at the time scale of the experiment, we reanalyzed data generated by *Gross et al., 2019* wherein cells were perturbed with IGF (37.5 pM), followed by a washout which allowed the cells to reach pre-stimulation nuclear FoxO levels, followed by a re-perturbation with the same amount of IGF. Nuclear FoxO response was measured at the single-cell level after 90 min with IGF exposure both these times. Since the response $x$ to the same input $u$ was measured twice in the same cell ($x_1$ and $x_2$), we could evaluate the intrinsic variability in response at the single-cell level. We then compared this intrinsic variability to the extrinsic cell state-dependent variability in the population.

To do so, we computed for each cell $\delta = x_1 - x_2$ the difference between the two responses. *Figure 2—figure supplement 7* shows the histogram $p(\delta)$ as computed from the data (pink) and the same computed from the model that was trained on the single-cell data (blue). We also computed $p(\delta_0)$, which represented the difference between responses of two different cells from the same population, shown for both data and the model.

As shown in *Figure 2—figure supplement 7*, the distribution $p(\delta)$ is significantly narrower than $p(\delta_0)$, suggesting that intracellular variability is significantly smaller than across-population variability and that cells' response to the same stimuli is quite conserved, especially when compared to responses in randomly picked pairs of cells. This shows that cell states and the corresponding response to extracellular perturbations are conserved, at least at the time scale of the experiment. Therefore, our estimates of cell-to-cell variability signaling fidelity are stable and reliable.

## 5. Model-predicted joint distributions $p_{CeeMI}(I, \chi)$ for several biochemical parameters
See *Figure 3—figure supplement 1*.

## 6. Examining the role of cell state dynamics on sensing ability
To elucidate the role of cell state dynamics, we built a simple model of ligand–receptor system (*Figure 4*). We define the system as follows:

$$\phi \overset{k_{\text{mprod}}}{\rightarrow} mRNA, \quad mRNA \overset{k_{\text{mdeg}}}{\rightarrow} \phi \tag{S27}$$

$$mRNA \overset{k_{\text{prod}}}{\rightarrow} mRNA + R \tag{S28}$$

$$R \overset{L \cdot k_{\text{bind}}}{\rightarrow} B, \quad B \overset{k_{\text{unbind}}}{\rightarrow} L + R \tag{S29}$$

$$R \overset{k_{\text{deg}}}{\rightarrow} \phi, \quad B \overset{k_{\text{deg}}}{\rightarrow} \phi \tag{S30}$$

Here all cells are assigned identical kinetic parameters. Instead, the state of the system is defined by the abundances of each molecule prior to ligand dose. Thus, each cell may be described by two state variables, $\vec{\theta} = \{R_0, mRNA_0\}$, where $mRNA_0$ is the initial mRNA levels and $R_0$ is the initial ligand-free receptor count. We tune the mRNA dynamics by changing mRNA production and degradation rates while keeping the average mRNA copy number constant. A relaxation of conditional responses requires both states have sufficiently high turnover, while we will modulate mRNA turnover, we will simply select a sufficiently high receptor turnover rate as described below.

For our simulations, we set $\frac{kmprod}{kmdeg} = 5$ a.u., $k_{prod} = 50$ a.u. $s^{-1}$, $k_{deg} = 0.5\ s^{-1}$, and $k_{unbind} = 1\ s^{-1}$. We also set $k_{\text{mdeg}} = \tau^{-1}$, where $\tau$ takes on five values, equally distributed in the log space, between $10^2$ and $10^4$ s. We select two distinct cells to study, $\vec{\theta}_a = \{300, 3\}$ and $\vec{\theta}_b = \{700, 7\}$. For each cell and mRNA turnover time scale of interest, we introduce 20 different doses equally distributed in log scale, leading to values of $Lk_{bind}$ ranging from $10^{-2}$ to $10^3\ (s^{-1})$. We ran 2500 Gillespie simulations for each condition to obtain approximate distributions of the response. We then fit these samples to gamma

distributions to obtain conditional responses from which the channel capacity mutual information can be calculated using an approach similar to the one described in Section 1.

## Acknowledgements

AG, HA, and PD were supported by NIGMS grant R35GM142547. The authors thank Andre Levchenko for useful discussions.

## Additional information

### Funding

| Funder | Grant reference number | Author |
|---|---|---|
| National Institute of General Medical Sciences | R35GM142547 | Andrew Goetz<br>Hoda Akl<br>Purushottam Dixit |

The funders had no role in study design, data collection and interpretation, or the decision to submit the work for publication.

### Author contributions

Andrew Goetz, Conceptualization, Software, Formal analysis, Validation, Investigation, Visualization, Methodology, Writing – original draft, Writing – review and editing; Hoda Akl, Conceptualization, Data curation, Software, Formal analysis, Validation, Investigation, Visualization, Methodology, Writing – original draft, Writing – review and editing; Purushottam Dixit, Conceptualization, Data curation, Funding acquisition, Investigation, Methodology, Writing – original draft, Project administration, Writing – review and editing

### Author ORCIDs

Purushottam Dixit (iD) https://orcid.org/0000-0003-3282-0866

Reviewer #2 (Public Review): https://doi.org/10.7554/eLife.87747.3.sa1
Reviewer #3 (Public Review): https://doi.org/10.7554/eLife.87747.3.sa2
Author Response https://doi.org/10.7554/eLife.87747.3.sa3

## Additional files

### Supplementary files
• MDAR checklist

### Data availability

The current manuscript is a computational study of previously collected data. No data have been generated for this manuscript. Scripts used to generate mutual information results as well the experimental data used to train our models can be obtained at: https://github.com/adgoetz186/Cell_signalling_information (copy archived at *Goetz, 2024*). Script for maximum entropy inference of cellular parameters can be obtained at: https://github.com/hodaakl/MaxEnt (copy archived at *Akl, 2024*).

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
