## [Editor Report · eLife assessment]

In this **valuable** article, the authors use an existing theoretical framework relying on information theory and maximum entropy inference in order to quantify how much information single cells can carry, taking into account their internal state. They reanalyze experimental data in this light. Despite some limitations of the data, the study **convincingly** highlights the difference between single-cell and population channel capacities. This result should be of interest to the quantitative biology community as it contributes to explaining why channel capacities are apparently low in cells.

---

## [Referee Report · Reviewer #2 (Public Review)]

In this paper the authors present an existing information theoretic framework to assess the ability of single cells to encode external signals sensed through membrane receptors. The main point is to distinguish actual noise in the signaling pathway from cell-cell variability, which could be due to differences in their phenotypic state, and to formalize this difference using information theory. After correcting for this cellular variability, the authors find that cells may encode more information than one would estimate from ignoring it, which is expected. The authors show this using simple models of different complexities, and also by analyzing an imaging dataset of the IGF/FoxO pathway.

I am only partially satisfied by the authors response. To me, the main question that was unanswered, while being at the core of the claim of the paper, was the magnitude of within-cell variability across repetitions of the stimulus.

This can only be done on the IGF/FoxO system because, as the authors acknowledge, the EGF/EGFR system does not have any data to support any claim about single-cell information that's not heavily informed by models, which assume by construction that this variability is small, naturally leading the desired conclusion.

The authors now measure within-cell, across-repetition variability (delta_0) for IGF/FoxO, but:

- they compare it to cell-to-cell variability, finding that it's smaller. That's good and that supports the main claim of the paper that single cells are more precise than a mean cell. However they don't show it in the paper, but only in the response.

- they also don't compare it to within-cell, within-stimulation variability across time. But this latter variability is what they (wrongly) used to estimate information, and still do in this revision. However I think this is approximated by the blue "simulation" violin plot in Reviewer Figure 2. The true variability is clearly larger than previously assumed. So it's strange that they conclude that "our estimates of cell-to-cell variability signaling fidelity are stable and reliable."

- they don't use this delta_0 variability to revise their estimate of the information accordingly.

- since variability is small compared to the differences between distinct stimulations, of which there are only 4, all information quantities they get are around 2 bits, which is not approaching the information capacity but merely a statement that the number of tested doses is small.

I strongly recommend that the authors actually report the figure they provided as Reviewer Figure 2 in the manuscript. In addition, they should not claim that the within-cell variability (captured by the variability across distinct presentations of the stimulus) is well captured by their initial estimate (based on the variance within a single presentation of the stimulus).

---

## [Referee Report · Reviewer #3 (Public Review)]

Goetz, Akl and Dixit investigated the heterogeneity in the fidelity of sensing the environment by individual cells in a population using computational modeling and analysis of experimental data for two important and well-studied mammalian signaling pathways: (insulin-like growth factor) IGF/FoxO and (epidermal growth factor) EFG/EFGR mammalian pathways. They quantified this heterogeneity using the conditional mutual information between the input (eg. level of IGF) and output (eg. level of FoxO in the nucleus), conditioned on the "state" variables which characterize the signaling pathway (such as abundances of key proteins, reaction rates, etc.) First, using a toy stochastic model of a receptor-ligand system - which constitutes the first step of both signaling pathways - they constructed the population average of the mutual information conditioned on the number of receptors and maximized over the input distribution and showed that it is always greater than or equal to the usual or "cell state agnostic" channel capacity. They constructed the probability distribution of cell state dependent mutual information for the two pathways, demonstrating agreement with experimental data in the case of the IGF/FoxO pathway using previously published data. Finally, for the IGF/FoxO pathway, they found the joint distribution of the cell state dependent mutual information and two experimentally accessible state variables: the response range of FoxO and total nuclear FoxO level prior to IGF stimulation. In both cases, the data approximately follow the contour lines of the joint distribution. Interestingly, high nuclear FoxO levels, and therefore lower associated noise in the number of output readout molecules, is not correlated with higher cell state dependent mutual information, as one might expect. This paper contributes to the vibrant body of work on information theoretic characterization of biochemical signaling pathways, using the distribution of cell state dependent mutual information as a metric to highlight the importance of heterogeneity in cell populations. The authors suggest that this metric can be used to infer "bottlenecks" in information transfer in signaling networks, where certain cell state variables have a lower joint distribution with the cell state dependent mutual information.

The utility of a metric based on the conditional mutual information to quantify fidelity of sensing and its heterogeneity (distribution) in a cell population is supported in the comparison with data. Some aspects of the analysis and claims in the main body of the paper and SI need to be clarified and extended.

Remaining Comments:

- I think Review Figure 2 which is currently in the SI would improve the main body of the paper if moved there. In that case, the discussion of this figure in the main text would have to address more than it currently does, namely "the same cell's FoxO responses to the same input were found to have significantly less variation compared to the variation within the population".

---

## [Author Response]

The following is the authors’ response to the original reviews.

**Public Reviews:**

**Reviewer #1 (Public Review):**
The manuscript by Goetz et al. takes a new perspective on sensory information processing in cells. In contrast to previous studies, which have used population data to build a response distribution and which estimate sensory information at about 1 bit, this work defines sensory information at the single cell level. To do so, the authors take two approaches. First, they estimate single cells' response distributions to various input levels from time-series data directly. Second, they infer these single-cell response distributions from the population data by assuming a biochemical model and extracting the cells' parameters with a maximum-entropy approach. In either case, they find, for two experimental examples, that single-cell sensory information is much higher than 1 bit, and that the reduction to 1 bit at the population level is due to the fact that cells' response functions are so different from each other. Finally, the authors identify examples of measurable cell properties that do or do not correlate with single-cell sensory information.The work brings an important and distinct new insight to a research direction that generated strong interest about a decade ago: measuring sensory information in cells and understanding why it is so low. The manuscript is clear, the results are compelling, and the conclusions are well supported by the findings. Several contributions should be of interest to the quantitative biology community (e.g., the demonstration that single cells' sensory information is considerably larger than previously implied, and the approach of inferring single-cell data from population data with the help of a model and a maximum-entropy assumption).

We thank the reviewer for the excellent summary of our research.

**Reviewer #2 (Public Review):**
In this paper the authors present an existing information theoretic framework to assess the ability of single cells to encode external signals sensed through membrane receptors.The main point is to distinguish actual noise in the signaling pathway from cell-cell variability, which could be due to differences in their phenotypic state, and to formalize this difference using information theory.After correcting for this cellular variability, the authors find that cells may encode more information than one would estimate from ignoring it, which is expected. The authors show this using simple models of different complexities, and also by analyzing an imaging dataset of the IGF/FoxO pathway.The implications of the work are limited because the analysed data is not rich enough to draw clear conclusions. Specifically,the authors do not distinguish what could be methodological noise inherent to microscopy techniques (segmentation etc), and actual intrinsic cell state. It's not clear that cell-cell variability in the analyzed dataset is not just a constant offset or normalization factor. Other authors (e.g. Gregor et al Cell 130, 153-164) have re-centered and re-normalized their data before further analysis, which is more or less equivalent to the idea of the conditional information in the sense that it aims to correct for this experimental noise.

We thank the reviewer for the comment. However, we do not believe our analysis is a consequence of normalization artifacts. Prior to modeling the single cell data, we removed well-dependent background fluorescence. This should take care of technical variation related to overall offsets in the data. We agree with the reviewer that background subtraction may not fully account for technical variability. For example, some of the cell-to-cell variability may potentially be ascribed to issues such as incorrect segmentation. Unfortunately, however, attempting to remove this technical variability through cell-specific normalization as suggested by the reviewer1 will diminish to a very large extent the true biological effects related to extensivity (cell size, total protein abundance). We note that these effects are a direct function of cell state-variables (see for example Cohen-Saidon et al.2 who use cell-state specific normalization to improve signaling fidelity). Therefore, an increase in mutual information after normalization does not only reflect removal of technical noise but also accounts for effect of cell state variables.

Nonetheless, as the reviewer suggested, we performed a cell-specific normalization wherein the mean nuclear FoxO levels in each cell (in the absence of IGF) were normalized to one. Then, for each ligand concentration, we collated FoxO response across all cells and computed the channel capacity corresponding to cell-state agnostic mutual information ICSA. As expected, ICSA increases from ∼0.9 bits to ∼1.3 bits when cell-specific normalization was performed (Author response image 1). However, this value is significantly lower than the average ∼1.95 of cell-state specific mutual information ⟨ICee⟩. Finally, we note that the cell specific normalization does not change the calculations of channel capacity at the single cell level as these calculations do not depend on linear transformations of the data (centering and normalization). Therefore, we do not think that our analysis of experimental data suffers from artifacts related to microscopy.

**Author response image 1. sa3fig1:** Author response image 1. Left: nuclear FoxO response averaged over all cells in the population across different ligand concentration. Right: nuclear FoxO response was first normalized at the single cell level and then averaged over all cells in the population across different ligand concentrations.

in the experiment, each condition is shown only once and sequentially. This means that the reproducibility of the response upon repeated exposures in a single cell was not tested, casting doubt on the estimate of the response fidelity (estimated as the variance over time in a single response).

The reviewer raises an excellent question about persistence of cell states. To verify that cell states are indeed conserved at the time scale of the experiment, we reanalyzed data generated by Gross et al.3 wherein cells were perturbed with IGF (37.5 pM), followed by a washout which allowed the cells to reach pre-stimulation nuclear FoxO levels, followed by a re-perturbation with the same amount of IGF. Nuclear FoxO response was measured at the single cell level after 90 minutes with IGF exposure both these times. Since the response x to the same input u was measured twice in the same cell (x1 and x2), we could evaluate the intrinsic variability in response at the single cell level. We then compared this intrinsic variability to the extrinsic cell-state dependent variability in the population.

To do so, we computed for each cell δ=x1-x2 the difference between the two responses. reviewer Figure 2 show the histogram p(δ) as computed from the data (pink) and the same computed from the model that was trained on the single cell data (blue). We also computed p(δ0) which represented the difference between responses of two different cells both from the data and from the model.

As we see in Author response image 2, the distribution p(δ) is significantly narrower than p(δ0) suggesting that intracellular variability is significantly smaller than across-population variability and that cells’ response to the same stimuli are quite conserved, especially when compared to responses in randomly picked pairs of cells. This shows that cell states and the corresponding response to extracellular perturbations are conserved, at least at the time scale of the experiment. Therefore, our estimates of cell-to-cell variability signaling fidelity are stable and reliable. We have now incorporated this discussion in the manuscript (lines 275-281).

**Author response image 2. sa3fig2:** Author response image 2. Left: Cells were treated with 37.5 pM of IGF for 90 minutes, washed out for 120 minutes and again treated with 37.5 pM of IGF. Nuclear FoxO was measured during the treatment and the washout. The distributions on the left show the difference in FoxO levels in single cells after the two 90 minutes IGF stimulations (pink: data, blue: model). Right: Distribution of difference in FoxO levels in two randomly picked cells after 90 minutes of exposure to 37.5 pM IGF.

another dataset on the EGF/EGFR pathway is analyzed, but no conclusion can be drawn from it because single-cell information cannot be directly estimated from it. The authors instead use a maximum-entropy Ansatz, which cannot be validated for lack of data.

We thank the reviewer for this comment. We agree with the reviewer that we have not verified our predictions for the EGF/EGFR pathway. That study was meant to show the potential generality of our analysis. We look forward to validating our predictions for the EGF/EGFR pathway in future studies.

**Reviewer #3 (Public Review):**
Goetz, Akl and Dixit investigated the heterogeneity in the fidelity of sensing the environment by individual cells in a population using computational modeling and analysis of experimental data for two important and well-studied mammalian signaling pathways: (insulin-like growth factor) IGF/FoxO and (epidermal growth factor) EFG/EFGR mammalian pathways. They quantified this heterogeneity using the conditional mutual information between the input (eg. level of IGF) and output (eg. level of FoxO in the nucleus), conditioned on the "state" variables which characterize the signaling pathway (such as abundances of key proteins, reaction rates, etc.) First, using a toy stochastic model of a receptor-ligand system - which constitutes the first step of both signaling pathways - they constructed the population average of the mutual information conditioned on the number of receptors and maximized over the input distribution and showed that it is always greater than or equal to the usual or "cell state agnostic" channel capacity. They constructed the probability distribution of cell state dependent mutual information for the two pathways, demonstrating agreement with experimental data in the case of the IGF/FoxO pathway using previously published data. Finally, for the IGF/FoxO pathway, they found the joint distribution of the cell state dependent mutual information and two experimentally accessible state variables: the response range of FoxO and total nuclear FoxO level prior to IGF stimulation. In both cases, the data approximately follow the contour lines of the joint distribution. Interestingly, high nuclear FoxO levels, and therefore lower associated noise in the number of output readout molecules, is not correlated with higher cell state dependent mutual information, as one might expect. This paper contributes to the vibrant body of work on information theoretic characterization of biochemical signaling pathways, using the distribution of cell state dependent mutual information as a metric to highlight the importance of heterogeneity in cell populations. The authors suggest that this metric can be used to infer "bottlenecks" in information transfer in signaling networks, where certain cell state variables have a lower joint distribution with the cell state dependent mutual information.The utility of a metric based on the conditional mutual information to quantify fidelity of sensing and its heterogeneity (distribution) in a cell population is supported in the comparison with data. Some aspects of the analysis and claims in the main body of the paper and SI need to be clarified and extended.1. The authors use their previously published (Ref. 32) maximum-entropy based method to extract the probability distribution of cell state variables, which is needed to construct their main result, namely p_CeeMI (I). The salient features of their method, and how it compares with other similar methods of parameter inference should be summarized in the section with this title. In SI 3.3, the Lagrangian, L, and Rm should be defined.

We thank the reviewer for the comment and apologize for the omission. We have now rewritten the manuscript to include references to previous reviews of works that infer probability distributions4 of cell state variables (lines 156-168). Notably, as we argued in our previous work5, no current method can efficiently estimate the joint distribution over parameters that is consistent with measured single cell data and models of signaling networks. Therefore, we could not use multiple approaches to infer parameter distributions. We have now expanded our discussion of the method in the supplementary information sections.

1. Throughout the text, the authors refer to "low" and "high" values of the channel capacity. For example, a value of 1-1.5 bits is claimed to be "low". The authors need to clarify the context in which this value is low: In some physically realistic cases, the signaling network may need to simply distinguish between the present or absence of a ligand, in which case this value would not be low.

We agree with the reviewer that small values of channel capacities might be sufficient for cells to carry out some tasks, in which case a low channel capacity does not necessarily indicate a network not performing its task. Indeed, how much information is needed for a specific task is a related but distinct question from how much information is provided though a signaling network. Both questions are essential to understand a cell's signaling behavior, with the former being far less easy to answer in a way which is generalizable. In contrast, the latter can be quantitatively answered using the analysis presented in our manuscript.

1. Related to (2), the authors should comment on why in Fig. 3A, I_Cee=3. Importantly, where does the fact that the network is able to distinguish between 23 ligand levels come from? Is this related to the choice (and binning) of the input ligand distribution (described in the SI)?

We thank the reviewer for the comment. The network can distinguish between all inputs used in the in silico experiment precisely because the noise at the cellular level is small enough that there is negligible overlap between single cell response distributions. Indeed, the mutual information will not increase with the number of equally spaced inputs in a sub-linear manner, especially when the input number is very high.

1. The authors should justify the choice of the gamma distribution in a number of cases (eg. distribution of ligand, distribution cell state parameters, such as number of receptors, receptor degradation rate, etc.).

We thank the reviewer for the comment. We note that previous works in protein abundances and gene expression levels (e.g. see6) have reported distributions with positive skews that can be fit well with gamma distributions or log-normal distributions. Moreover, many stochastic models of protein abundance levels and signaling networks are also known to result in abundances that are distributed according to a negative binomial distribution, the discrete counterpart of gamma distribution. Therefore, we chose Gamma distributions in our study. We have now clarified this point in the Supplementary Information. At the same time, gamma distribution only serves as a regularization for the finite data and in principle, our analysis and conclusion do not depend on choice of gamma distribution for abundances of proteins, ligands, and cell parameters.

1. Referring to SI Section 2, it is stated that the probability of the response (receptor binding occupancy) conditioned on the input ligand concentration and number of receptors is a Poisson distribution. Indeed this is nicely demonstrated in Fig. S2. Therefore it is the coefficient of variation (std/mean) that decreases with increasing R0, not the noise (which is strictly the standard deviation) as stated in the paper.

We thank the reviewer of the comment. We have now corrected our text.

1. In addition to explicitly stating what the input (IGF level) and the output (nuclear GFP-tagged FoxO level) are, it would be helpful if it is also stated what is the vector of state variables, theta, corresponding to the schematic diagram in Fig. 2C.

We thank the reviewer of the comment. We have now corrected our text in the supplementary material as well as the main text (Figure 2 caption).

1. Related to Fig. 2C, the statement in the caption: "Phosphorylated Akt leads to phosphorylation of FoxO which effectively shuttles it out of the nucleus." needs clarification: From the figure, it appears that pFoxO does not cross the nuclear membrane, in which case it would be less confusing to say that phosphorylation prevents reentry of FoxO into the nucleus.

We thank the reviewer of the comment. We have now corrected our text (Figure 2 caption).

1. The explanations for Fig. 2D, E and insets are sparse and therefore not clear. The authors should expand on what is meant by model and experimental I(theta). What is CC input dose? Also in Fig. 2E, the overlap between the blue and pink histograms means that the value of the blue histogram for the final bin - and therefore agreement or lack thereof with the experimental result - is not visible. Also, the significance of the values 3.25 bits and 3 bits in these plots should be discussed in connection with the input distributions.

We thank the reviewer of the comment. We have now corrected our text (Figure 2 caption and lines 249-251).

1. While the joint distribution of the cell state dependent mutual information and various biochemical parameters is given in Fig. S7, there is no explanation of what these results mean, either in the SI or main text. Related to this, while a central claim of the work is that establishing this joint distribution will allow determination of cell state variables that differentiate between high and low fidelity sensing, this claim would be stronger with more discussion of Figs. 3 and S7. The related central claim that cell state dependent mutual information leads to higher fidelity sensing at the population level would be made stronger if it can be demonstrated that in the limit of rapidly varying cell state variables, the I_CSA is retrieved.

We thank the reviewer for this excellent comment. We have now added more discussion about interpreting the correlation between cell state variables and cell-state specific mutual information (lines 294-306). We also appreciate the suggestion about a toy model calculation to show that dynamics of cell state variables affects cell state specific mutual information. We have now performed a simple calculation to show how dynamics of cell state variables affects cells’ sensing ability (lines 325-363). Specifically, we constructed a model of a receptor binding to the ligand wherein the receptor levels themselves changed over time through a slow process of gene expression (Author response image 3, main text Figure 4). In this model, the timescales of fluctuations of ligand-free receptors on the cell surface can be tuned by speeding up/slowing down the degradation rate of the corresponding mRNA while keeping the total amount of steady state mRNA constant. As shown in Author response image 3, the dependence of cell-specific mutual information on cell state variable diminishes when the time scale of change of cell state variables is fast.

**Author response image 3. sa3fig3:** Author response image 3. Cell state dynamics governs cell state conditioned mutual information. A. In a simple stochastic model, receptor mRNA is produced at a constant rate from the DNA and the translated into ligand-free receptors. The number of ligand-bound receptors after a short exposure to ligands is considered the output. B. A schematic showing dynamics of receptor numbers when mRNA dynamics are slower compared to signaling time scales. C. Conditioning on receptor numbers leads to differing abilities in sensing the environment when the time scale of mRNA dynamics τ is slow. In contrast, when the mRNA dynamics are fast (large τ-1), conditioning on cell state variables does not lead to difference in sensing abilities.

**Reviewer #1 (Recommendations For The Authors):**
My major concerns are mainly conceptual, as described below. With proper attention to these concerns, I feel that this manuscript could be a good candidate for the eLife community.Major concerns:1. The manuscript convincingly demonstrates that cells good sensors after all, and that heterogeneity makes their input-output functions different from each other. This raises the question of what happens downstream of sensing. For single-celled organisms, where it may be natural to define behavioral consequences at the single-cell level, it may very well be relevant that single-cell information is high, even if cells respond differently to the environment. But for cells in multicellular organisms, like those studied here, I imagine that most behavioral consequences of sensing occur at the multicellular level. Thus, many cells' responses are combined into a larger response. Because their responses are different, their high-information individual responses may combine into a low-information collective response. In fact, one could argue that a decent indicator of the fidelity of this collective response is indeed the population-level information measure estimated in previous works. Thus, a fundamental question that the authors must address is: what is the ultimate utility of reliable, but heterogeneous, responses for a multicellular system? This question has an important bearing for the relevance of their findings.

We thank the reviewer for this thought-provoking comment. We agree that the fidelity with which cells sense their environment, especially those in multicellular organisms, may not always need to be very high. We speculate that when the biological function of a collection of cells can be expressed as an average over the response of individual cells; high-information but heterogeneous cells can be considered equivalent to low-information homogeneous cells. An example of such a function is population differentiation to maintain relative proportions of different cell types in a tissue or producing a certain amount of extracellular enzyme.

In contrast, we believe that when the biological function involves collective action, spatial patterning, or temporal memory, the difference between reliable but heterogeneous population and unreliable homogeneous population will become significant. We plan to explore this topic in future studies.

1. The authors demonstrate that the agreement is good between their inference approach and the direct estimation of response distributions from single-cell time series data. In fact, the agreement is so good that it raises the question of why one would need the inference approach at all. Is it because single-cell time series data is not always available? Is that why the authors used it for one example and not the other? The validation is an asset, but I imagine that the inference approach is complicated and may make assumptions that are not always true. Thus, its utility and appropriate use must be clarified.

We thank the reviewer for the comment. As the reviewer correctly pointed out, live cell imaging data is not always available and has limited scope. Specifically, optical resolution limits measurements of multiple targets. Moreover, typical live cell measurements measure total abundance or localization and not post-translational modification (phosphorylation, methylation, etc.) which are crucial to signaling dynamics. The most readily available single cell data such those measured using single cell RNA sequencing, immunofluorescence, or flow cytometry are necessarily snapshots. Therefore, computational models that can connect underlying signaling networks to snapshot data become essential when imputing single cell trajectories. In addition, the modeling also allows us to identify network parameters that correlate most strongly with cellular heterogeneity. We have now clarified this point in the manuscript (lines 366-380).

Minor comments:1. I would point out that the maximum values in the single-cell mutual information distributions (Fig 2D and E) correspond to log2 of the number of inputs levels, corresponding to perfect distinguishability of each of the equally-weighted input states. It is clear that many of the mutual information values cluster toward this maximum, and it would help readers to point out why.

We thank the reviewer for the comment. We have now included a discussion about the skew in the distribution in the text (lines 251-260).

1. Line 216 references Fig 2C for the EGF/EGFR pathway, but Fig 2C shows the FoxO pathway. In fact, I did not see a schematic of the EGF/EGFR pathway. It may be helpful to include one, and for completeness perhaps also one for the toy model, and organize the figures accordingly.

We thank the reviewer for the comment. We did not include three separate schematics because the schematics of the EGF/EGFR model and the toy model are subsets of the schematic of the IGF/FoxO model. We have now clarified this point in the manuscript (Figure 2 caption).

**Reviewer #2 (Recommendations For The Authors):**
the simple model of Fig. 2A would gain from a small cartoon explaining the model and its parameters.

We thank the reviewer for the comment. We did not include a schematic for the toy model as it is a subset of the schematic of the IGF/FoxO model. The schematic of the toy model is included in the supplementary information.

L should be called u, and B should be called x, to be consistent with the rest of the notations in the paper.

We have decided to keep the notation originally presented in the manuscript.

legend of 2E and D should be clarified. "CC input dose" is cryptic. The x axis is the input dose, the y axis is its distribution at the argmax of I. CC is the max of I, not its argmax. Likewise "I" in the legend for the colors should not be used to describe the insets, which are input distributions.

We have now changed this in the manuscript.

the data analysis of the IGF/FoxO pathway should be explained in the main text, not the SI. Otherwise it's impossible to understand how one arrives at, or how to intepret, figure 2E, which is central to the paper. For instance the fact that p(x|u,theta) is assumed to be Gaussian, and how the variance and mean are estimated from the actual data is very important to understand the significance of the results.

While we have added more details in the manuscript in various places, for the sake of brevity and clarity, we have decided to keep the details of the calculations in the supplementary materials.

there's no Method's section. Most of the paper's theoretical work is hidden in the SI, while it should be described in the methods.

We thank the review of the comment. However, we believe that adding a methods section will break the narrative of the paper. The methods are described in detail in the supplementary materials with sufficient detail to reproduce our results. Additionally, we also provide a link to the github page that has all scripts related to the manuscript.

PS: please submit a PDF of the SI for review, so that people can read it on any platform (as opposed to a word document, especially with equations)

We have now done this.

**Reviewer #3 (Recommendations For The Authors):**
1. Subplots in Fig. 1, inset in Fig. 3 are not legible due to small font.

We have now increased the font.

1. Mean absolute error in Fig. S5 and relative error in related text should be clarified.

We have now clarified this in the manuscript.

1. Acronyms (MACO, MERIDIAN) should be defined.

We have now made these changes.

References

1. Gregor T, Tank DW, Wieschaus EF, Bialek W. Probing the limits to positional information. Cell. 2007;130(1):153-64. doi: 10.1016/j.cell.2007.05.025. PubMed PMID: WOS:000248587000018.

2. Cohen-Saidon C, Cohen AA, Sigal A, Liron Y, Alon U. Dynamics and Variability of ERK2 Response to EGF in Individual Living Cells. Mol Cell. 2009;36(5):885-93. doi: 10.1016/j.molcel.2009.11.025. PubMed PMID: WOS:000272965400020.

3. Gross SM, Dane MA, Bucher E, Heiser LM. Individual Cells Can Resolve Variations in Stimulus Intensity along the IGF-PI3K-AKT Signaling Axis. Cell Syst. 2019;9(6):580-8 e4.

4. Loos C H, J. Mathematical modeling of variability in intracellular signaling. Current Opinion in Systems Biology. 2019;16:17-24.

5. Dixit PD, Lyashenko E, Niepel M, Vitkup D. Maximum Entropy Framework for Predictive Inference of Cell Population Heterogeneity and Responses in Signaling Networks. Cell Syst. 2020;10(2):204-12 e8.

6. Taniguchi Y, Choi PJ, Li GW, Chen H, Babu M, Hearn J, Emili A, Xie XS. Quantifying *E. coli* proteome and transcriptome with single-molecule sensitivity in single cells. Science. 2010;329(5991):533-8. doi: 10.1126/science.1188308. PubMed PMID: 20671182; PMCID: PMC2922915.